# A Cross-Sectional Study: Systematic Quantification of Chemerin in Human Cerebrospinal Fluid

**DOI:** 10.3390/biomedicines12112508

**Published:** 2024-11-01

**Authors:** Alexandra Höpfinger, Manuel Behrendt, Andreas Schmid, Thomas Karrasch, Andreas Schäffler, Martin Berghoff

**Affiliations:** 1Department of Internal Medicine III, University of Giessen, Klinikstr. 33, 35392 Giessen, Germany; manuel.s.behrendt@med.uni-giessen.de (M.B.); andreas.schmid@innere.med.uni-giessen.de (A.S.); thomas.karrasch@innere.med.uni-giessen.de (T.K.); andreas.schaeffler@innere.med.uni-giessen.de (A.S.); 2Department of Neurology, University of Giessen, Klinikstr. 33, 35392 Giessen, Germany; martin.berghoff@klinikum-msp.de

**Keywords:** adipoflammation, adipokine, chemerin, fat–brain axis, metaflammation

## Abstract

Background: Dysregulation of adipokines is considered a key mechanism of chronic inflammation in metabolic syndrome. Some adipokines affect food intake by crossing the blood/brain barrier. The adipokine chemerin is associated with metabolic syndrome, cardiovascular diseases and immune response. Little is known about chemerin’s presence in cerebrospinal fluid (CSF) and its ability to cross the blood/CSF barrier. Methods: We quantified chemerin levels in paired serum and CSF samples of 390 patients with different neurological diagnoses via enzyme-linked immunosorbent assay (ELISA). Correlation analyses of serum and CSF chemerin levels with anthropometric, serum and CSF routine parameters were performed. Results: Overweight patients exhibited higher chemerin levels in serum and CSF. Chemerin CSF levels were higher in men. Chemerin levels in serum were associated with BMI (body mass index) and CRP (C-reactive protein). Chemerin levels in CSF were associated with age. Neurological diseases affected chemerin levels in CSF. The chemerin CSF/serum ratio was calculated as 96.3 ± 36.8 × 10^−3^ for the first time. Conclusions: Our data present a basis for the development of standard values for chemerin quantities in CSF. CSF chemerin levels are differentially regulated in neurological diseases and affected by BMI and sex. Chemerin is able to cross the blood/CSF barrier under physiological and pathophysiological conditions.

## 1. Introduction

Obesity is a well-known risk factor for cardiovascular diseases, type 2 diabetes mellitus, dyslipidemia and metabolic syndrome [1]. It is thought that obesity-related diseases are caused by dysregulated adipokine production [2]. Increased infiltration of macrophages in adipose tissue and a shift to proinflammatory adipokines/cytokines lead to chronic systemic inflammation in obese patients [2], referred to as metaflammation [3]. Furthermore, obesity is assumed to play a role in chronic inflammatory diseases of the central nervous system [4] and neurodegenerative diseases [5]. Via the so-called fat–brain axis, adipokines might affect brain metabolism, brain atrophy, cognitive decline and neuronal inflammation [6].

Incretins influence eating habits by crossing the blood/CSF (cerebrospinal fluid) barrier [7]. Therapeutic approaches like GLP-1 (glucagon like peptide) analogues exploit this effect [8]. They are able to reduce food intake by mediating satiety [8]. Similar effects are known or expected for different adipokines. Most prominent, leptin was one of the first adipokines to be discovered in CSF [9] and it is transported via the blood/CSF barrier [10]. Mutations in the leptin gene or in leptin receptor gene lead to uncontrolled food intake and early childhood obesity [11] since leptin acts as a satiety signal [12]. Several adipokines have previously been described in CSF [13], each with completely different permeability described by the CSF/serum ratio. An important adipokine in the focus of current research is chemerin. Chemerin is thought to be crucial in physiological and pathophysiological processes in obesity, such as adipogenesis, insulin sensitivity and innate immune response [14]. Furthermore, chemerin plays a vital role in vascular inflammation, blood pressure modulation and angiogenesis [15]. At the site of inflammation, chemerin acts as an important chemoattractant for macrophages and dendritic cells [16]. Known chemerin receptors are G-coupled receptors inducing secretion of proinflammatory cytokines like IL (interleukin)-6 and TNF (tumor necrosis factor)-α [17]. An intracellular pathway involving STAT (signal transducer and activator of transcription) 3 signaling has been proposed [18]. Chemerin serum levels are elevated in obesity and can be significantly reduced by weight-loss [19] and exercise training [20]. Interestingly, chemerin serum levels might be influenced by diet since they are reduced by oral lipid ingestion [21]. In rodents, intracerebral chemerin infusion promotes food intake [22]. Furthermore, expression of the chemerin receptor was found in hippocampal tissue [23]. This finding motivates the hypothesis, that chemerin might modulate food intake via central pathways if it is able to cross the blood/brain barrier under physiological and pathophysiological conditions. Surprisingly, studies on chemerin CSF levels in humans are sparse. There are no studies available on chemerin levels in CSF in patients without neurological diseases. Therefore, it is unknown if chemerin is able to cross the blood/CSF barrier so far. In a small study cohort of 31 patients suffering from brain tumors or miscellaneous central nervous system (CNS) diseases, chemerin levels were quantified in CSF with concentrations ranging from 2.9 to 6.3 ng/mL [24]. Furthermore, this study reported significantly up-regulated chemerin mRNA in grade III and IV gliomas [25]. In murine multiple sclerosis (MS) models, the hypothesis was raised that there is a potential link between disease severity and chemerin expression in the CNS [26]. Tomalka-Kochanowska et al. found that obese MS patients had higher chemerin plasma levels than lean MS patients or lean controls [27]. But, unfortunately, their study was lacking CSF samples. Studies on chemerin levels in CSF in human MS patients are not available so far. As chemerin is a modulator of the innate immune system [16], it might be involved in central infectious diseases. This issue has not been investigated until now.

To the best of our knowledge, there are no systematic studies in large cohorts available investigating paired chemerin CSF and serum levels in humans. Therefore, little is known about chemerin’s ability to cross the blood/brain barrier under physiological and pathophysiological conditions in vivo in humans. Therefore, the aim of the present study was to quantify paired chemerin serum and CSF levels in a large study population (n = 390) of patients undergoing neurological evaluation, provide standard values for chemerin CSF levels in patients with/without neurological diseases, calculate chemerin CSF/serum ratio for the description of permeability and identify relevant correlations between serum and CSF chemerin levels, anthropometric parameters and routine laboratory parameters in a large study cohort to gain insight into chemerin’s ability to cross the blood/brain barrier under physiological and pathophysiological conditions. 

## 2. Materials and Methods

### 2.1. Study Cohort

In total, 390 patients undergoing diagnostic or therapeutic lumbar puncture in the neurological department of University Hospital Giessen, Germany, from 2016 to 2023 were enrolled in this study. Paired CSF and serum samples were obtained under sterile conditions and stored at −20 °C. Patients gave informed consent to participate in this study, which was approved by the local ethical committee (registration note AZ 13/16). There was no prior selection of patients other than the need of neurological evaluation indicated by a board-certified neurologist. 

Age and anthropometric parameters such as height, body weight and BMI (body mass index) were recorded. Patients were classified into BMI subgroups of overweight patients (BMI ≥ 25 kg/m^2^) and non-overweight patients (BMI < 25 kg/m^2^). Serum routine parameters like creatinine, urea, glucose, LDH (lactate dehydrogenase), transaminases, triglycerides, LDL (low density lipoprotein) cholesterol, HDL (high density lipoprotein) cholesterol, total cholesterol, CRP (C-reactive protein) and TSH (thyroid-stimulating hormone), as well as leukocyte cell count and HbA1c, were analyzed by standard and routine methods at the Institute of Clinical Chemistry and Laboratory Medicine, University Hospital Giessen, Germany. Metabolic diseases such as diabetes mellitus, dyslipidemia and hypertension were documented. Medication (antidiabetic, contraceptives, statins, antihypertensive drugs) and smoking habits were recorded. CSF routine parameters such as cell count, total protein, Ig (immunoglobulins) M, A, G, oligoclonal bands and albumin were analyzed by standard routine methods in the Neurochemical Laboratory Giessen University Hospital, Germany. Patients were categorized into subgroups regarding their CSF cell count (<5 cells/µL; ≥5 cells/µL). Furthermore, concerning their grade of blood/brain barrier dysfunction, patients were classified into subgroups (normal, slight, moderate, severe dysfunction of the blood/brain barrier) at the Neurochemical Laboratory Giessen University Hospital, Germany, by standard methods, e.g., CSF/serum albumin ratio and cell count in CSF. A board-certified neurologist provided the neurological diagnoses. Patients were categorized into subgroups regarding their neurological diagnosis: control group without neurological disease (n = 170) after finishing the clinical work up, MS (n = 68), infectious disease of the central nervous system (ID) (n = 30), epilepsy (n = 50), cerebrovascular disease (CVD) (e.g., patients suffering from stroke in differing cerebral areas and transient cerebral ischemic attacks) (n = 46) and pseudotumor cerebri (PC) (=idiopathic intracranial hypertension) (n = 26). For subgroup analyses, patients matched for age, sex and BMI were selected from the control group. Here, we chose patients from the control group with an CSF/serum albumin ratio × 10^−3^ < (4 + age/15) × 10^−3^ as recommended in literature [28].

### 2.2. Quantification of Serum and CSF Chemerin Levels

Serum and CSF chemerin levels were investigated by enzyme-linked immunosorbent assay (ELISA) (DuoSet ELISA Development Kits, Biotechne, Wiesbaden, Germany). Measurements were performed in technical duplicates and an intra-duplicate deviation of <20% was accepted. The detection range of the ELISA kit was 31.2–2000 pg/mL. Standard curve was made according to the instructions from the selling company: a series of dilutions with factor 2. (e.g., 2000, 1000, 500, etc.) was conducted. The dilution buffer 1% bovine serum albumin (BSA) in phosphate-buffered saline (PBS) was used for serum and CSF samples.

### 2.3. Statistical Analyses

For all statistical analyses, a statistical software package (IBM SPSS Statistic 28.0, USA) was used. Mean values ± standard deviation (SD) were calculated and minimum and maximum values were indicated. Since the analyzed data do not follow normal distribution and independent samples were compared, nonparametric tests were applied. Mean values of numerical parameters were compared by the Mann–Whitney U test (for 2 unrelated samples) or by the Kruskal–Wallis test (for more than 2 unrelated parameters). Bivariate correlation analyses were performed using the Spearman-rho test. The Chi-Quadrat-Test was applied for identification of confounding variables. *p* values of <0.05 (two tailed) were considered as statistically significant. 

Some figures are displayed as boxplots. Boxplots display the median and lower and upper interquartile range. Whiskers mark minimal and maximal values. 

Power analysis was performed for applying G*Power 3.1.9.7 (free software provided by University of Düsseldorf) in order to confirm sample sizes to be sufficient for providing a statistical power of 0.8 (effect size d = 0.5) in two-way comparisons.

## 3. Results

### 3.1. Study Cohort

In the present study, 390 (161 men and 229 women) patients from the Department of Neurology, University Hospital Giessen, Germany, were included. Table 1 summarizes detailed characteristics of the entire study population. Anthropometric and physiological parameters are displayed by mean ± SD and range. The mean age was 45.6 ± 17.9 years, and 41.3% of patients were not overweight, whereas 58.7% were overweight (BMI ≥ 25 kg/m^2^). Concerning their neurological diagnosis, patients were divided into six main subgroups, as shown in Table 1 and Figure 1. One hundred seventy individuals (43.6% of the whole cohort) had no evidence of any neurological disease. Among the 220 classified neurological patients, MS (n = 68, 17.4%), ID such as meningitis or encephalitis (n = 30, 7.7%), epilepsy (n = 50, 12.8%), CVD (n = 46, 11.8%) and PC (n = 26, 6.7%) could be diagnosed.

### 3.2. Quantification of Chemerin Levels Among the Entire Study Cohort

Chemerin levels could be successfully quantified in serum and CSF of all 390 patients (Table 1). In serum, chemerin levels ranged about ~10-fold from 92.34 to 864.27 ng/mL. The mean value was 348.02 ± 118.54 ng/mL. In CSF, chemerin levels were substantially lower (~1/10) and ranged from 8.48 to 69.51 ng/mL with a mean value of 31.02 ± 9.80 ng/mL. In overweight patients, chemerin serum levels were significantly higher than in normal weight patients (*p* < 0.001) (Figure 2A). Similar to this observation, chemerin CSF levels were significantly elevated in overweight patients (*p* = 0.012) but to a lesser extent (Figure 2B). Male patients exhibited significantly higher chemerin CSF levels than female patients (*p* < 0.001) (Figure 2D), whereas serum chemerin levels did not differ regarding the sex of the patients (Figure 2C). We were able to exclude BMI as a confounding variable of the observed sexual dimorphism (Appendix A); however, men exhibited a significantly higher prevalence of impaired blood/brain barrier dysfunction than women (*p* < 0.001) (Figure 3). All correlation analyses performed for the entire study cohort are displayed in Appendix A.

We investigated chemerin serum levels in subgroup analysis regarding neurological diagnoses (Figure 4A). In the control group, chemerin serum levels ranged from 150.22 to 775.28 ng/mL (mean: 352.37 ± 112.42 ng/mL). Patients suffering from MS had significantly lower chemerin levels in serum 297.71 ± 113.26 ng/mL than controls (*p* = 0.005). Patients suffering from CVD (mean: 395.96 ± 154.77 ng/mL) (*p* = 0.002) and PC (mean: 385.18 ± 99.60 ng/mL) (*p* = 0.003) had significantly higher chemerin serum levels compared to MS, whereas mean values of patients with either ID or epilepsy did not significantly differ in chemerin serum levels from the control group or MS patients.

In CSF, chemerin levels did not differ between controls (mean: 31.06 ± 8.25 ng/mL) and neurological patients, whereas patients with either MS (mean: 27.80 ± 8.69 ng/mL) (vs. CVD: *p* = 0.001; vs. epilepsy: *p* = 0.010) or infectious diseases (mean: 26.74 ± 10.67 ng/mL) (vs. CVD: *p* = 0.005; vs. epilepsy: *p* = 0.025) exhibited lower CSF chemerin levels when compared to patients suffering from CVD (mean: 35.06 ± 10.87 ng/mL) or epilepsy (mean: 34.77 ± 12.48 ng/mL) (Figure 4B).

Since we observed differing chemerin levels in CSF associated with specific neurological diagnoses, we performed subgroup analyses concerning markers of CSF inflammation and chemerin levels in CSF. Patients with oligoclonal bands exhibited significantly lower chemerin levels in CSF (*p* = 0.001) (Figure 5A). Furthermore, in patients with cell count above normal in CSF (≥5/µL), lower chemerin levels in CSF were detected (*p* < 0.001) (Figure 5B). Patients with slightly dysfunctional blood/brain barrier had higher chemerin CSF levels (*p* = 0.008), whereas moderate or severe dysfunction of blood/brain barrier did not significantly affect chemerin CSF levels (Figure 5C).

### 3.3. Chemerin Levels in Patients Without Evidence of a Neurological Disease

In patients without neurological disease, extensive correlation analyses with serum and CSF chemerin levels were performed (Appendix A). Chemerin serum levels correlated positively with BMI (*p* < 0.001; rho = +0.414; n = 170) (Figure 6A) and CRP levels (*p* < 0.001; rho = +0.570; n = 166) (Figure 6B). Importantly, chemerin levels in serum and CSF showed a strong positive correlation (*p* = 0.008; rho = +0.201; n = 170) (Figure 6C) and the specific CSF/serum-ratio for chemerin was calculated as 96.3 ± 36.8 × 10^−3^ for the first time. In CSF, chemerin levels correlated positively with age (*p* < 0.001; rho = +0.427; n = 170) (Figure 6D) and CSF/serum albumin ratio (*p* < 0.001; rho = +0.381; n = 164) (Figure 6E), an observation which is probably explained by an increased permeability of the blood/brain barrier by increasing age. CSF/serum chemerin and CSF/serum albumin ratio correlated positively as well (*p* = 0.021; rho = +0.180; n = 164) (Figure 6F). Thus, chemerin is permeable to the brain and increased albumin permeability (e.g., caused by disease) leads to increased chemerin permeability. Age, BMI, and CRP could be ruled out as confounding variables (Appendix A). Interestingly, the positive correlation between chemerin serum levels and BMI and correlations with chemerin CSF levels were gender dependent and mainly found in the female sub-cohort only (Appendix A). All correlation analyses performed for the sub-cohort of patients without neurological diseases are displayed in Appendix A.

### 3.4. Chemerin Levels in Patients with Neurological Diseases Compared to Matched Control Groups

In our study cohort, 68 (17.4%) patients suffered from MS. We selected a control cohort of 68 subjects matched by age, BMI and sex from the individuals without a neurological disease and with a CSF/serum albumin ratio of <(4 + age/15) × 10^−3^ (as recommended [28]). Chemerin serum levels in MS patients ranged from 92.34 to 707.25 ng/mL with a mean value of 297.71 ± 113.26 ng/mL and were significantly lower than in matched controls (*p* = 0.014). In matched controls, chemerin serum levels exhibited a mean value of 338.86 ± 101.58 ng/mL. CSF chemerin levels ranged from 10.70 to 46.92 ng/mL with a mean value of 27.80 ± 8.69 ng/mL in MS patients and did not significantly differ from matched controls (30.26 ± 8.01 ng/mL). Table 2A shows characteristics of the matched sub-cohorts (MS and matched controls). 

Thirty patients had infectious diseases such as meningitis or encephalitis. We matched controls without neurological diseases regarding sex, age and BMI as described above. Patients with infectious diseases had lower chemerin levels in CSF controls (*p* = 0.011). Furthermore, chemerin CSF/serum ratio was lower in these patients (*p* = 0.046). In Table 2B, we display characteristics of patients suffering from infectious diseases and matched controls.

Fifty patients in the study cohort had epilepsy. Patients were compared with matched controls without a neurological disease as described above. CSF chemerin levels were higher in patients with epilepsy compared to matched controls (*p* = 0.037), whereas no differences in chemerin serum levels were observed. Table 2C shows detailed characteristics of epilepsy patients and controls.

As described above, we matched patients with and without neurological diseases. In Appendix A, characteristics of patients suffering from CVD (Appendix A) and PC (Appendix A) and matching controls (matched for sex, age and BMI) are depicted. Mean values of chemerin serum or CSF levels did not differ significantly between those neurological subgroups and matched controls. 

## 4. Discussion

The present study investigated circulating and CSF chemerin levels in paired samples from a large clinical cohort. Comprised patients suffered from various neurological diseases such as MS, ID, epilepsy, CVD and PC, as well as a control group of individuals without evidence of neurological diseases. Chemerin represents an adipokine and chemoattractant protein involved in a plethora of immunological processes [16], carbohydrate metabolism [14] and development of cardiovascular diseases [15]. Chemerin levels in serum are elevated in patients with obesity and nonalcoholic fatty liver disease [29]. Interestingly, intrathecal chemerin infusion leads to increased food intake [22]. Therefore, chemerin levels in CSF in humans are of major interest regarding regulation of feeding behavior by chemerin, hypothetically acting along a putative fat–brain axis of this chemokine.

We were able to quantify serum and CSF chemerin in all 390 samples. Consistent with previous findings in a smaller study cohort (n = 31) [24], chemerin levels in CSF are around 10% of serum levels. Serum and CSF chemerin levels exhibited a strong correlation in the entire study cohort, as well as in the sub-cohort of individuals without neurological diseases. To our knowledge, no comparably large studies have investigated chemerin levels in CSF so far. Therefore, our study provides a solid basis in order to evaluate standard values for chemerin CSF levels in specific pathological contexts. Moreover, we present a carefully evaluated CSF/serum ratio for chemerin that can be used for future studies and evaluations of other hormones and adipokines. In clinical practice, CSF/serum ratios are of high relevance. Especially CSF/serum ratios of albumin and immunoglobulins are used for diagnostic purposes [28]. Increased ratios are either caused by impaired blood/brain barrier function or by an increased autochthonous production in the CNS. Reduced CSF/serum ratios can be due to increased serum levels and stable blood/brain barrier permeability, e.g., if the substance is actively transported via the blood/brain barrier independent of serum concentrations. The observed chemerin CSF/serum ratio of 96.3 × 10^−3^ is comparable to the C1q/TNF-related protein^−3^ (CTRP-3) CSF/serum ratio (110 × 10^−3^) [30], remarkably higher compared to the leptin CSF/serum ratio 3.9 × 10^−3^ [31] (3.8 × 10^−3^ [32]) or resistin CSF/serum ratio (4.4 × 10^−3^) [13] but, interestingly, lower than the Meteorin-Like Protein (METRNL) CSF/serum ratio (1400 × 10^−3^ [33]). These adipokines were recently described in CSF, suggesting them as putative mediators of the fat–brain axis. However, for most adipokines, the mechanisms by which they cross the blood/brain barrier are unknown so far. 

Within the total study cohort, we observed significantly higher CSF chemerin quantities in male patients when compared to women. Of particular interest, we found a sexual dimorphism of this protein in CSF. Previous studies detected significant differences in circulating chemerin, some reporting higher levels in male [34] or in female individuals [21]. In contrast, we did not observe a significant difference in serum chemerin levels regarding sex. Furthermore, since no difference in CSF chemerin levels between men and women was detected within the sub-cohort of individuals with negative neurological diagnosis, it appears reasonable to assume that substantial effects of neurological pathologies and/or anthropometric parameters such as BMI might underly the dimorphism in CSF detected within the total study cohort. Interestingly, men were significantly more likely to have a dysfunctional blood/brain barrier than women in the present cohort. Thus, higher prevalence of blood/brain barrier dysfunction might substantially contribute to elevated CSF chemerin levels within the male subgroup, potentially alongside further neurological disease-related factors such as inflammatory environment. In our retrospective study, as well as in real life, gender is not equally distributed among neurological diseases. For example, infectious diseases with impaired blood/brain barrier occur in similar proportion in male and female patients. Patients suffering from pseudotumor cerebri, who rarely show impaired blood/brain barrier function, are more likely to be female. These gender-related differences regarding neurological pathologies may contribute to the significant difference of chemerin levels in CSF by gender. A previous study has shown that testosterone inhibits chemerin secretion in murine adipocytes in vitro [21]. Therefore, increased chemerin in CSF in male patients is more likely caused by surrounding factors (e.g., dysfunctional blood/brain barrier, disease-related factors) than direct regulation via androgens. Nonetheless, chemerin receptors have been found in Leydig cells of the male reproductive tract and chemerin was suggested as a regulator in male gonadal steroidogenesis [35]. Future studies should investigate effects of central chemerin on reproductive functions.

In the current cohort, we observed a positive correlation of systemic chemerin levels with BMI (and CRP) and elevated chemerin levels in overweight patients consistent with previous studies [36]. Chemerin is strongly expressed in adipocytes and induced during adipocyte differentiation, reaching its apex in mature adipocytes [36]. Chemerin appears to play an autocrine/paracrine role in white adipose tissue [37]. It is able to affect recruitment of macrophages in white adipose tissue via inhibition of matrix metalloproteinase 3 and chemokines via NFkB signaling [37]. Endogenous chemerin can increase chemerin activity via autocrine positive feedback [37]. 

For the first time in the literature, we were able to show an analogous correlation of BMI and chemerin in serum in CSF, with significantly elevated chemerin CSF levels in overweight individuals. Different hypotheses can be drawn from this observation: increased chemerin levels in serum could lead to proportionally increased chemerin levels in CSF. Furthermore, the blood/brain barrier could be more permeable for chemerin in obese patients. Increased autochthone chemerin production in the central nervous system in obese patients also cannot be excluded due to our study design. Future studies should be designed to test these hypotheses.

As systemic chemerin levels are associated with several key aspects of the metabolic syndrome [36], elevated chemerin quantities in CSF represent an intriguing issue. Chemerin receptors have been described in the hypothalamus [38], but regulation of chemerin in CSF and mechanisms of chemerin crossing the blood/brain barrier have not been investigated so far. For the best-researched adipokine leptin, a saturable, specific, temperature-dependent receptor was found at the human blood/brain barrier [39]. In our study, obese patients had elevated CSF/serum ratios. This might support the hypothesis that chemerin levels in CSF depend, at least in part, on chemerin serum concentrations. Future studies should investigate molecular mechanisms of chemerin crossing the blood/brain barrier under physiological and pathophysiological conditions in vivo. 

CSF chemerin levels are rising with increasing age, similar to blood/brain barrier permeability [28]. Therefore, we cannot exclude that the observed correlation between chemerin CSF levels and age might be biased by a dysfunctional blood/brain barrier. Of note, chemerin levels in CSF were elevated in patients with mild dysfunction of the blood/brain barrier but not in patients with moderate or severe barrier dysfunction. However, the latter finding might be due to the limited case numbers for moderate/severe barrier dysfunction. The association of CSF chemerin quantities with stages of increasing barrier dysfunction should therefore be verified and studied in more detail in future clinical approaches focusing on this issue. Regarding the CNS, chemerin expression has previously been reported in the hippocampus and in the cortex of mice [40]. Therefore, it appears reasonable to assume autochthone chemerin production in the CNS as an additional substantial source of chemerin in CSF. 

Chemerin is an important immune-modulating factor in systemic autoimmune diseases such as systemic sclerosis [41] or rheumatoid arthritis [42]. Therefore, chemerin might also be relevant in other autoimmune diseases such as MS. Studies in murine MS models revealed a potential link between disease severity and chemerin expression in the CNS [26]. Chemerin CSF levels in MS human patients have not been extensively investigated so far and are of great interest due to the promising observations in the MS murine models [26]. In our study cohort, comparative analysis of pathological subgroups revealed significantly lower serum chemerin levels in MS patients when compared to neurologically healthy individuals, both with and without matching for age, sex and BMI, whereas chemerin levels in CSF were unchanged in MS compared to matched controls. Nonetheless, CSF chemerin levels were significantly lower in MS patients when compared to patients suffering from epilepsy or CVD. Consistent with these results, patients positive for oligoclonal bands exhibited lower chemerin levels in CSF. In contrast to the present findings, a previous study reported chemerin serum quantities to be elevated in MS in association with overweight and obesity [27]. Overall, the current data imply a slight reduction of circulating chemerin in MS. In a murine model of autoimmune encephalomyelitis, the chemerin receptor CMKLR1 (chemokine-like receptor-1) is expressed in microglial cells and CNS-infiltrating myeloid dendritic cells [26]. In vitro, chemerin is able to trigger β-arrestin-2 association with CMKLR1 and induce cell migration of these CMKLR1^+^ cells [43]. Antagonizing these chemerin effects suppressed CNS autoimmune inflammation in a murine model of autoimmune encephalomyelitis [43]. Moreover, chemerin receptor-deficient mice exhibited less severe disease phenotypes [26]. Therefore, chemerin appears to play a pivotal role in development of MS, at least in mice. Future studies should focus on the role of chemerin in patients suffering from MS. 

Chemerin is a well-known modulator in the innate immune system and increases the release of proinflammatory cytokines like tumor necrosis factor (TNF)-α or interleukin (IL)-6 [44] via G protein-coupled receptors [17]. Furthermore, chemerin is able to limit bacterial infection [45]. Meanwhile, its potential role in the context of infectious CNS disease has not been investigated so far. Of particular interest, comparative analysis with matched controls revealed decreased CSF chemerin quantities in patients suffering from infectious diseases of the CNS while serum levels did not significantly differ. Furthermore, the CSF/serum ratio was significantly lower in patients suffering from infectious diseases. Consistent with this finding, the subset of patients with elevated cell counts in CSF (≥ 5/µL) exhibited significantly lower chemerin levels, and CSF chemerin levels were negatively correlated with cell count in general. Interestingly, in the entire study cohort, as well as in the controls without neurological diseases, CRP levels correlated positively with chemerin levels in serum. However, neither serum nor CSF chemerin levels were associated with systemic inflammatory parameters such as CRP in the sub-cohort of patients suffering from infectious CNS diseases. Taken together, these findings might indicate infection-driven down-regulation of either autochthonous chemerin expression specifically in the CNS or of chemerin permeability through the blood/CSF barrier. Nonetheless, chemerin expression is increased in an inflammatory environment by TNF-α [46], yet proteolytic procession of chemerin was found at the site of inflammation [47] and might result in decreased chemerin levels in CSF in inflammatory diseases. Future studies should test these hypotheses and elucidate the relevance and mechanisms of reduced chemerin levels in infectious CNS diseases.

While there is substantial knowledge of systemic chemerin regulation and involvement in viral and bacterial infections [48,49], data on effects on CSF chemerin levels have not yet been available. Within the present cohort, infectious diseases are mainly represented by meningitis and encephalitis. The current data might therefore shed light upon a novel aspect of these pathologies and should motivate further research on the involvement of chemerin in CNS infection and inflammation.

We found a slight yet significant elevation of chemerin levels in CSF in patients suffering from CVD compared to MS or infectious diseases in the entire study cohort. In a murine stroke model, recombinant chemerin was reported to exert protective effects on neuronal and blood/brain barrier damage [50]. Furthermore, chemerin reduces microglial inflammatory response and neuronal apoptosis [51]. Therefore, an upregulation of chemerin in CVD patients can be speculated to reduce neuronal and blood/brain barrier damage. Future studies should investigate chemerin’s impact on neuronal damage in CVD patients. 

In the pathogenesis of epilepsy, reactive microglia play a major role in neuroinflammation, phagocytosis, and remodeling of the epileptic brain microenvironment [52]. Interestingly, chemerin promotes the migration of microglia [40] and therefore might be relevant in the pathogenesis of epilepsy. In our study cohort, 50 patients suffered from epilepsy. In CSF, chemerin levels were significantly elevated, whereas serum quantities did not differ from those of matched controls. Therefore, chemerin levels in CSF might represent a new diagnostic parameter in patients suffering from epilepsy. Future studies should investigate chemerin CSF levels in larger cohorts of epileptic patients, with further focus on disease severity. Interestingly, increased serum chemerin levels were also observed in idiopathic epilepsy and chemerin concentrations were associated with seizure severity, as was reported by a previous study including 50 children [53]. However, unfortunately, this study was lacking CSF samples [53]. Expression of chemerin receptor was found in hippocampal tissue after status epilepticus [23], but corresponding function of chemerin receptor in hippocampal tissue has not been investigated so far. Underlying mechanisms of chemerin in epilepsy remain to be elucidated and the current data provide a solid basis for future mechanistic studies focusing on this intriguing issue.

## 5. Limitations

In the present study, we analyzed serum and CSF samples of 390 patients in a retrospective cross-sectional approach. Therefore, there was no prior selection of patients participating in this study. This might be seen as a strength or a weakness of this study. For example, no individual researcher’s bias is to be expected in the selection of patients. But we were not able to analyze specific routine parameters like glucose in CSF retrospectively. Some parameters were not measured in all patients, e.g., parameters of the lipid metabolism, and we are not able to complete missing data. Patient history was assessed by documentation; however, some relevant yet unavailable information might be missing due to the period of time between collecting data and conducting the study. A previous study has shown that chemerin serum levels are reduced after oral lipid ingestion [21]. Due to the retrospective nature of the present study, we have no information about patients eating habits prior to sampling of serum and CSF. Therefore, short-term effects of ingested lipids cannot be excluded. Moreover, future research should focus on mechanistic studies to explore the mechanism of chemerin crossing the blood/brain barrier. 

## 6. Conclusions

For the first time in the literature, we were able to present chemerin levels in paired samples in serum and CSF in a large and well-characterized study cohort. With this study, we provide a basis for standard values of chemerin in CSF in patients with and without neurological diseases. As a novel aspect, we demonstrated the pleiotropic adipokine chemerin in the context of relevant neurological diseases. Future studies should address decreased chemerin levels in MS/ID and increased levels in epilepsy. The permeability of chemerin to the brain is within the range of CTRP-3, higher than leptin and resistin and lower than METRNL. 

## Figures and Tables

**Figure 1 biomedicines-12-02508-f001:**
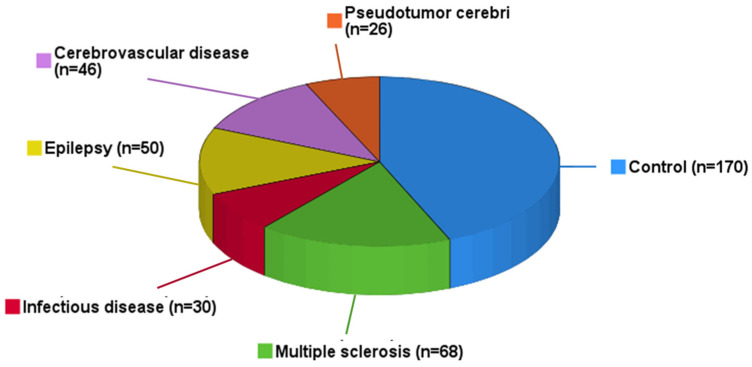
Distribution of neurological diagnoses among study participants: 390 patients were divided into subgroups regarding their final neurological diagnosis. The diagnosis was provided by a board-certified neurologist. The figure displays prevalence of each neurological subgroup.

**Figure 2 biomedicines-12-02508-f002:**
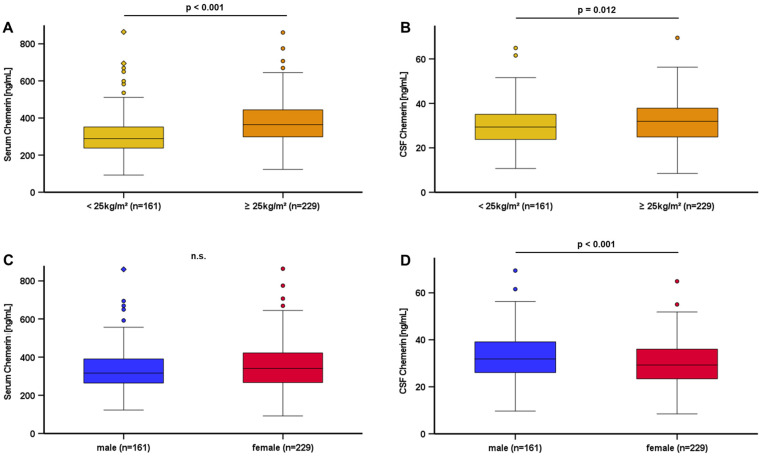
Chemerin levels in serum and CSF by BMI and gender: Chemerin levels in serum (**A**) and CSF (**B**) were elevated in overweight patients. There was no sexual dimorphism in chemerin serum quantities (**C**). Chemerin levels in CSF were higher in men (**D**). Chemerin levels were quantified by ELISA. The Mann–Whitney U test was applied for calculation. A *p* value for statistical significance was defined as *p* < 0.05 without correction for multiple comparisons. Data are displayed in boxplots. Boxplots display the median, lower and upper interquartile range. Whiskers mark minimum and maximum. CSF: cerebrospinal fluid.

**Figure 3 biomedicines-12-02508-f003:**
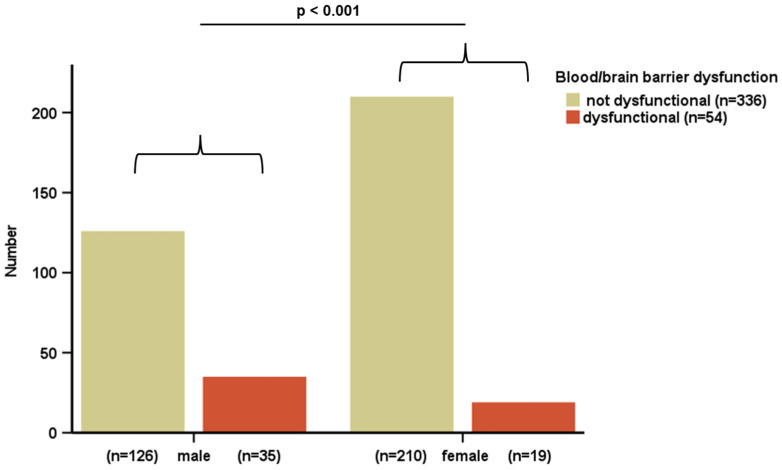
Prevalence of blood/brain barrier dysfunction by gender: All patients were divided into two subgroups (no blood/brain barrier dysfunction; blood/brain barrier dysfunction). Within these subgroups, a division regarding gender was performed. The prevalence of dysfunctional blood/brain barrier was compared. Significantly more men exhibited signs of impaired blood/brain barrier dysfunction. Chi-Quadrat-Test was applied for calculation of *p* values and statistical significance (*p* < 0.05). Blood/brain barrier dysfunction was classified by the Neurochemical Laboratory of the University Hospital Giessen, Germany, as no blood/brain barrier dysfunction (n = 336) and blood/brain barrier dysfunction (n = 54).

**Figure 4 biomedicines-12-02508-f004:**
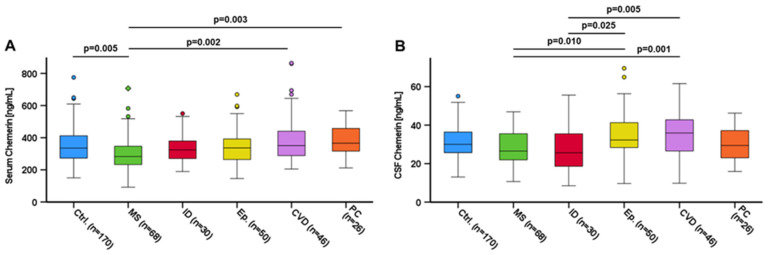
Chemerin levels in serum by neurological diagnosis (**A**) and chemerin levels in CSF by neurological diagnosis (**B**). Chemerin levels were quantified by ELISA. Kruskal–Wallis test was applied for calculation of *p* values and statistical significance (*p* < 0.05) applying Bonferroni correction. Data are displayed in boxplots. Boxplots display the median, lower and upper interquartile range. Whiskers mark minimum and maximum. CSF: cerebrospinal fluid. Ctrl.: control; MS: multiple sclerosis; ID: infectious disease; Ep.: epilepsy; CVD: cerebrovascular disease; PC: pseudotumor cerebri.

**Figure 5 biomedicines-12-02508-f005:**
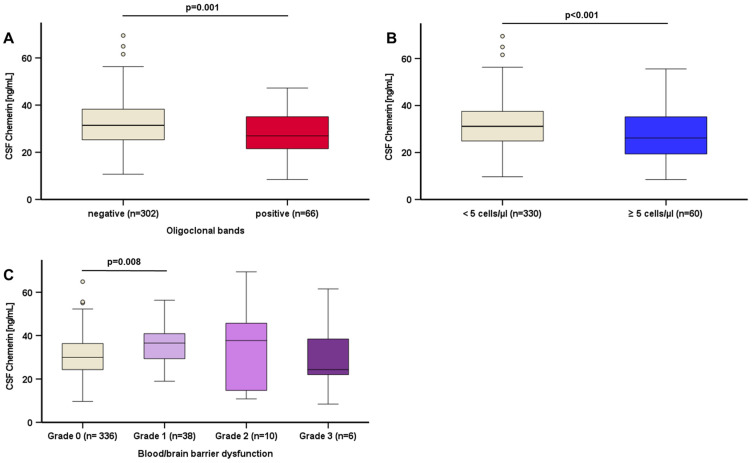
Chemerin levels in CSF regarding inflammatory markers: Chemerin levels in CSF by oligoclonal band status (**A**). Chemerin Levels in CSF by cell count in CSF (**B**). Chemerin levels in CSF by grade of blood/brain barrier dysfunction (**C**). Chemerin levels were quantified by ELISA. Status of oligoclonal band, cell count in CSF and grading of blood/brain barrier dysfunction were provided by the Neurochemical Laboratory Giessen University Hospital, Germany. The Mann–Whitney U test (2 subgroups) and the Kruskal–Wallis test (>2 subgroups, applying Bonferroni correction) were applied for calculation of *p* values and statistical significance (*p* < 0.05). Data are displayed in boxplots. Boxplots display the median and lower and upper interquartile range. Whiskers mark minimum and maximum. CSF: cerebrospinal fluid.

**Figure 6 biomedicines-12-02508-f006:**
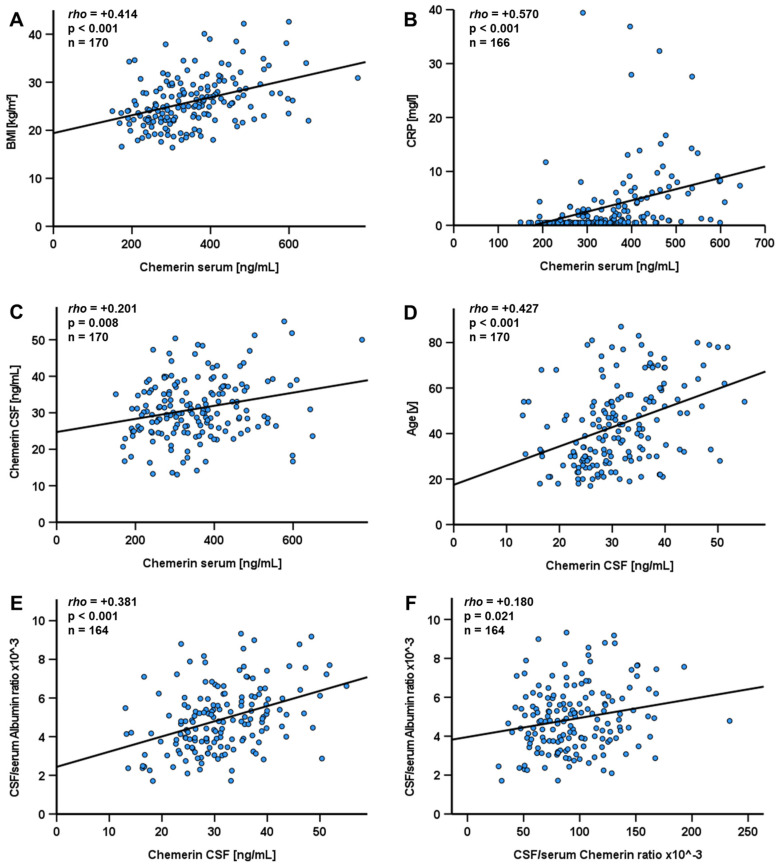
Correlation analyses with chemerin serum and CSF levels in patients without indications of neurological disease. Correlation between chemerin levels in serum and BMI (**A**). Correlation between chemerin levels in serum and CRP (**B**). Correlation between chemerin levels in serum and chemerin CSF levels (**C**). Correlation between chemerin levels in CSF and age (**D**). Correlation between chemerin levels in CSF and CSF/serum albumin ratio (**E**). Correlation between CSF/serum chemerin ratio and CSF/serum albumin ratio (**F**). Chemerin levels were quantified by ELISA. The Spearman-rho test was applied for calculation of *p* values and statistical significance (*p* < 0.05). Statistical outliers > 3× standard deviation were excluded from analysis. BMI: body mass index; CRP: C-reactive protein; CSF: cerebrospinal fluid.

**Table 1 biomedicines-12-02508-t001:** Characteristics of the whole study cohort.

Study Population (n = 390)	
Men, n (%)	161 (41.3)
Women, n (%)	229 (58.7)
**Anthropometric parameters**	
Age [y]	45.6 ± 17.9 [17–87]
Weight [kg]	79.9 ± 20.2 [44.0–190.0]
Height [cm]	171.3 ± 8.9 [146–194]
Mean BMI (kg/m^2^)	27.19 ± 6.31 [16.4–62.0]
BMI < 25 kg/m^2^, n (%)	161 (41.3)
BMI ≥ 25 kg/m^2^, n (%)	229 (58.7)
**Serum parameters**	
*Carbohydrate metabolism*	
Random plasma glucose [mg/dL]	108.97 ± 35.43 [62–308]
HbA1c [%]	5.72 ± 0.74 [2.0–8.3]
Diabetes mellitus type 2, n (%)	27 (6.9)
*Lipoprotein metabolism*	
Total cholesterol [mg/dL]	192 ± 48 [82–335]
LDL cholesterol [mg/dL]	126 ± 47 [33–287]
HDL cholesterol [mg/dL]	50 ± 16 [14–102]
Triglycerides [mg/dL]	149 ± 91 [35–559]
*Inflammation*	
CRP [mg/L]	6.09 ± 18.02 [0.50–284.11]
Leukocytes [giga/L]	7.71 ± 2.73 [1.7–22.5]
Total protein [g/L]	70.65 ± 6.13 [52–85]
Total albumin [g/L]	42.42 ± 4.76 [22.8–55.3]
GOT [U/L]	22.22 ± 13.31 [8–124]
GPT [U/L]	28.06 ± 30.15 [7–445]
Bilirubin [mg/dL]	0.66 ± 0.66 [0.1–9.0]
Creatinine [mg/dL]	0.82 ± 0.66 [0.4–13.3]
Urea [mg/dL]	27.89 ± 11.49 [9–129]
LDH [U/L]	191.68 ± 44.44 [92–375]
**CSF parameters**	
Cell count [cells/µL]	11.77 ± 55.35 [0–608]
Total protein [g/L]	1.09 ± 13.30 [0.118–263]
Albumin [g/L]	0.26 ± 0.17 [0.06–1.56]
Lactate [mmol/L]	1.77 ± 0.41 [1.00–3.44]
IgG [g/L]	0.037 ± 0.048 [0.007–0.733]
IgA [g/L]	0.005 ± 0.022 [0.00027–0.34300]
IgM [g/L]	0.001 ± 0.005 [0.00000–0.059700]
CSF/serum albumin ratio × 10^−3^	6.18 ± 4.25 [1.68–40.52]
**Adipokines**	
Chemerin in serum [ng/mL]	348.02 ± 118.54 [92.34–864.27]
Chemerin in CSF [ng/mL]	31.02 ± 9.80 [8.48–69.51]
CSF/serum chemerin ratio × 10^−3^	96.31 ± 36.78 [21.37–233.583]
**Neurological diseases/Clinical subgroups**	
Control, n (%)	170 (43.6)
Multiple sclerosis, n (%)	68 (17.4)
Infectious disease, n (%)	30 (7.7)
Epilepsy, n (%)	50 (12.8)
Cerebrovascular disease, n (%)	46 (11.8)
Pseudotumor cerebri, n (%)	26 (6.7)

Characteristics of the whole study cohort: Anthropometric and laboratory parameters in serum and CSF are displayed. Mean values, standard deviation and range are shown. Absolute numbers and percentages are given for classified variables. BMI: body mass index; LDL: low-density lipoprotein; HDL: high-density lipoprotein; CRP: C-reactive protein; GOT: glutamic oxaloacetic transaminase; GPT: glutamic-pyruvic transaminase; CSF: cerebrospinal fluid; LDH: lactate dehydrogenase; Ig: immunoglobulin.

**Table 2 biomedicines-12-02508-t002:** Chemerin levels in patients suffering from neurological diseases compared to matched control groups.

(A)	MS (n = 68)	Matched Controls (n = 68)
Chemerin serum [ng/mL]	297.71 ± 113.26 [92.34–707.25] *	338.86 ± 101.58 [168.65–599.64] *
Chemerin CSF [ng/mL]	27.80 ± 8.69 [10.70–46.92]	30.26 ± 8.01 [13.27–50.39]
Chemerin CSF/serum ratio × 10^−3^	102.76 ± 40.17 [38.73–221.18]	95.83 ± 32.74 [27.79–167.31]
**(B)**	**ID (n = 30)**	**Matched Controls (n = 30)**
Chemerin serum [ng/mL]	335.06 ± 90.89 [189.66–550.76]	364.57 ± 141.82 [150.22–775.28]
Chemerin CSF [ng/mL]	26.74 ± 10.67 [8.48–55.59] *	33.48 ± 8.28 [14.17–51.84] *
Chemerin CSF/serum ratio × 10^−3^	81.92 ± 31.05 [33.40–159.10] *	101.83 ± 38.69 [39.82–233.58] *
**(C)**	**Epilepsy (n = 50)**	**Matched Controls (n = 50)**
Chemerin serum [ng/mL]	346.03 ± 109.06 [146.26–669.26]	354.20 ± 108.30 [168.65–649.70]
Chemerin CSF [ng/mL]	34.77 ± 12.48 [9.69–69.51] *	30.41 ± 8.84 [13.07–51.25] *
Chemerin CSF/serum ratio × 10^−3^	106.08 ± 38.49 [21.37–226.71]	93.81 ± 37.73 [27.79–193.04]

Chemerin levels in patients suffering from neurological diseases compared to matched control groups. Chemerin levels in serum and CSF and chemerin CSF/serum ratio in MS (A), ID (B) and epilepsy (C) patients and matched controls are displayed. Mean values, standard deviation and range are displayed. MS: multiple sclerosis; ID: infectious disease; CSF: cerebrospinal fluid. *p* values were calculated by Mann–Whitney U test, *: *p* < 0.05.

## Data Availability

The data that support the findings of this study are available from the corresponding author upon reasonable request.

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
