# Peer review of "A Cross-Sectional Study: Systematic Quantification of Chemerin in Human Cerebrospinal Fluid"

_biomedicines, 2024, doi:10.3390/biomedicines12112508_

Round 1
Reviewer 1 Report
Comments and Suggestions for Authors
• The design of the study should be mentioned in the title. E.g. cross sectional.
• The detailed affiliation should be added.
• In the abstract the term correlated should replaced by associated.
• The abstract should be shortened not to exceed 200 words and its structure should meet the MDPI standards.
• The keywords should be changed and enriched, and should not be similar to terms already mentioned in the title. Moreover should be ordered alphabetically.
• In the Introduction section the aim should be presented as one paragraph, and should be followed by a hypothesis of the study.
• A power analysis should be conducted to justify the suitableness of the sample size.
• Authors should conduct more sophisticated statistical analysis, as associations followed by regression models including confounders (age, gender, BMI etc. in the same model).
• The quality of tables should be improved, to meet MDPI standards.
• The Discussion section should rewritten to include the following subsections:
1) Main findings of the study and their comparison with previously published literature on the topic
2) The clinical implication of the findings
3) The strength and limitations should be clearly mentioned
4) The new directions for future research to conduct on the topic
Author Response
Reviewer 1
- The design of the study should be mentioned in the title. E.g. cross sectional.
Thank you for this important recommendation. We added the term “cross-sectional” in the title. (l. 2, p. 1)
- The detailed affiliation should be added.
Affiliations are mentioned below authors:
“ 1 Department of Internal Medicine III, University of Giessen, Klinikstr. 33, 35392 Giessen, Germany
2 Department of Neurology, University of Giessen, Klinikstr. 33, 35392 Giessen, Germany”
(p.1, l. 6-7)
- In the abstract the term correlated should replaced by associated.
We thank the reviewer for this suggestion and replaced the term correlated by associated (p. 1, ll. 19-20)
- The abstract should be shortened not to exceed 200 words and its structure should meet the MDPI standards.
We want to thank the editor for this valuable comment. We shortened and structured the abstract as recommended. (p. 1, ll. 10-24)
- The keywords should be changed and enriched, and should not be similar to terms already mentioned in the title. Moreover should be ordered alphabetically.
We thank the reviewer for this hint. We changed and enriched the keywords and ordered them alphabetically. (p. 1, l. 25)
- In the Introduction section the aim should be presented as one paragraph, and should be followed by a hypothesis of the study.
Thank you for this important recommendation. We rephrased the aim of the study. (p. 2, ll. 78-84)
- A power analysis should be conducted to justify the suitableness of the sample size.
Thank you for this recommendation. As was suggested by the reviewer, power analysis was performed applying G*Power 3.1.9.7 (free software provided by University of Düsseldorf). For all comparisons of chemerin serum and CSF concentrations between patient subgroups, the group sizes being available within the clinical cohort and applied to the statistical tests were confirmed to be sufficient in order to provide a statistical power of 0.8 in two-way comparisons. We added this in the methods section (p. 3, ll. 144-146)
- Authors should conduct more sophisticated statistical analysis, as associations followed by regression models including confounders (age, gender, BMI etc. in the same model).
Thank you for this helpful suggestion. We conducted further statistical analysis to exclude confounders in our most import findings in correlation analyses. Interestingly, gender appears to have a relevant effect on chemerin in CSF. We now mention these findings in the results section and added the results in the supplementary file. (p. 9, ll. 281-285, supplementary table 3)
- The quality of tables should be improved, to meet MDPI standards.
Thank you for this comment. We added the heading above the tables. If you suggest any further changes in the layout of the tables, please let us know. To our knowledge, the tables meet the MDPI standards
- The Discussion section should rewritten to include the following subsections:
1) Main findings of the study and their comparison with previously published literature on the topic
2) The clinical implication of the findings
3) The strength and limitations should be clearly mentioned
4) The new directions for future research to conduct on the topic
We thank the reviewer for this important recommendation. We elaborated the discussion section as suggested e.g. (p. 12, ll. 354-360, ll. 381-394, p. 13, ll. 405-420, p. 13, ll. 414-420; p. 14, ll. 448-456
Furthermore, we added a section “limitations” in our manuscript. (p. 15, ll. 511-524)
In the present study, we analyzed serum and CSF samples of 390 patients in a retrospective cross-sectional approach. Therefore, there was no prior selection of pa-tients participating in this study. This might be seen as a strength or a weakness of this study. For example, no individual researche’s bias is to be expected in the selection of patients. But we were not able to analyze specific routine parameters like glucose in CSF retrospec-tively. Some parameters were not measured in all patients e.g. parameters of the lipid metabolism and we are not able to complete missing data. Patients’ history was as-sessed by documentation, however, some relevant yet unavailable information might be missing due to the period of time between collecting data and conducting the study. A previous study has shown that chemerin serum levels are reduced after oral lipid ingestion [21]. Due to the retrospective nature of the present study, we have no infor-mation about patients eating habits prior to sampling of serum and CSF. Therefore, short-term effects of ingested lipids cannot be excluded. Moreover, future research should focus on mechanistical studies to explore the mechanism of chemerin crossing the blood/brain barrier.” (p.15, ll. 510-524)
Reviewer 2 Report
Comments and Suggestions for Authors
Review Report of manuscript ID Number” Biomedicines- 3224902”
The research article entitled “Systematic quantification of chemerin in human cerebrospinal fluid” submitted for publication in “Biomedicnes” with manuscript ID Number” Biomedicines-3224902” has good healthy and comprehensive results and the manuscript is well designed and presented and contains all the necessary experiments to prove his hypothesis.
My general comments on the manuscript are incorporated as follows:
The title should be capitalized correctly: “Systematic Quantification of Chemerin in Human Cerebrospinal Fluid”
In the abstract there are several grammatical and textual mistakes such as
The unit for the CSF/serum ratio should be consistently written as “× 10⁻³” instead of “x10-3”
There are inconsistencies in the number of decimal places used such as “96.3+36.8 x10-3” should be “96.3 ± 36.8 × 10⁻³”.
The term “enzyme-linked-immunosorbent assay” should be hyphenated correctly as “enzyme-linked immunosorbent assay (ELISA)”.
The abbreviations for body mass index (BMI) and C-reactive protein (CRP) should be defined when first mentioned.
The phrase “chemerins ability” should be “chemerin’s ability”
Some sentences are long and complex, making them difficult to read such as “Dysregulation of adipokines and inflammation of adipose tissue are considered as key mechanisms of chronic inflammation in metabolic syndrome (metaflammation)” could be split into two sentences for clarity.
There are missing commas in several places, such as after introductory phrases (e.g., “Interestingly, chemerin serum levels might be influenced by diet since they are reduced by oral lipid ingestion”).
In the abstract “Data on chemerin in human cerebrospinal fluid (CSF) is sparse. Little is known about chemerins ability to cross the blood/CSF barrier under physiological and pathophysiological conditions” could be revised to “Data on chemerin in human cerebrospinal fluid (CSF) is sparse, and its ability to cross the blood/CSF barrier under various conditions is not well understood”.
Some statements in the abstract are too predetermine such as “Our data present a solid basis for the development of standard values for chemerin quantities in CSF” are too strong given the study’s limitations.
In the introduction:
The sentence “Obesity-related diseases are thought to be caused by dysregulated adipokine production” could be revised to “It is thought that obesity-related diseases are caused by dysregulated adipokine production”.
The sentence “In rodents, intracerebral chemerin infusion promotes food intake and expression of chemerin receptor was found in hippocampal tissue” could be split into two sentences for clarity.
There are missing commas in several places, such as after introductory phrases (e.g., “Interestingly, chemerin serum levels might be influenced by diet since they are reduced by oral lipid ingestion”).
The units“BMI ≥ 25kg/m2” should have a space between the number and the unit: “BMI ≥ 25 kg/m²”.
Some references are incomplete or incorrectly formatted. For example, “doi:10.1038/nm0596-589” should be formatted consistently with other references.
Results:
It is mentioned that the Mann-Whitney U-test and Kruskal-Wallis tests were used for statistical analysis. However, it does not provide sufficient detail on the assumptions of these tests.
The results section also presents a lot of data in text form, which can be organized into more tables and figures to summarize key findings to improve readability.
The study mentions excluding BMI as a confounding variable for sex differences in chemerin levels, but it does not address other potential confounders comprehensively.
In Table 1
The unit for BMI should be written as “kg/m²” instead of “kg/m2”.
The unit for cell count should be “cells/µL” instead of “uL”.
The CSF/serum albumin ratio should be written as “CSF/serum albumin ratio × 10⁻³” instead of “CSF/serum Albumin ratio x10-3”.
There are inconsistencies in the number of decimal places used. For example, “BMI (kg/m²)” is given as “27.19 + 6.31” while “Total protein (g/L)” is “70.65 + 6.13”. Consistency should be maintained throughout the table.
The ranges for some parameters are not clearly defined. For example, “Total albumin (g/L)” has a range of “[22.8-414.0]”, which seems unusually high and might indicate an error.
The abbreviations for low-density lipoprotein and high-density lipoprotein should be defined in the table legend or footnotes for clarity.
Parameters should be grouped logically. For example, anthropometric parameters (age, weight, height, BMI) should be grouped together, followed by serum parameters, and then CSF parameters.
Adding subgroup headings (e.g., “Anthropometric Parameters”, “Serum Parameters”, “CSF Parameters”) would improve readability.
Anthropometric Parameters should be listed first, followed by serum parameters, and then CSF parameters. This logical flow helps in understanding the data better.
The neurological subgroups should be listed at the end of the table to maintain a clear distinction between general parameters and specific subgroups.
The table legend should provide a detailed description of the parameters, including units and any abbreviations used. For example, “LDL: low-density lipoprotein; HDL: high-density lipoprotein; CRP: C-reactive protein; GOT: glutamic oxaloacetic transaminase; GPT: glutamic-pyruvic transaminase; CSF: cerebrospinal fluid; LDH: lactate dehydrogenase; Ig: immunoglobulin”.
Footnotes should be used to clarify any unusual values or ranges. For example, if the range for “Total albumin (g/L)” is correct, a footnote should explain why it is so high.
Figure 1: Neurological Diseases
The figure lacks clear labels for each subgroup. Ensure that each subgroup is clearly labeled and distinguishable. The spelling of infectious disease is incorrect.
The legend is missing or insufficient. A detailed legend explaining the subgroups and any abbreviations used is necessary.
The title should be more descriptive. For example, “Distribution of Neurological Diagnoses Among Study Participants”.
How were the subgroups determined? Were there any specific criteria?
Figure 2: Chemerin Levels in Serum and CSF
Ensure that units are consistently used and clearly indicated on the axes.
The p-values are indicated, but the method of calculation should be mentioned in the figure legend.
The titles for each subplot (A, B, C, D) should be more descriptive. For example, “Chemerin Levels in Serum by BMI” instead of just “Serum Chemerin [ng/ml]”.
Were the p-values adjusted for multiple comparisons?
Figure 3: Blood/Brain Barrier Dysfunction
The figure is not very clear in distinguishing between the groups. Use different colors or patterns to improve clarity.
Ensure that units are indicated where applicable.
The title should be more descriptive. For example, “Prevalence of Blood/Brain Barrier Dysfunction by Gender”. A detailed legend explaining the groups and any abbreviations used is necessary.
How was blood/brain barrier dysfunction assessed?
Figure 4: Chemerin Levels by Neurological Diagnosis
Ensure that error bars are included to indicate variability.
The method of calculating p-values should be mentioned in the figure legend.
The titles for each subplot (A, B) should be more descriptive. For example, “Chemerin Levels in Serum by Neurological Diagnosis”.
Were the comparisons adjusted for potential confounders?
Please include error bars to indicate variability.
Figure 5: Chemerin Levels and CSF Inflammation Markers
The figure is not very clear in distinguishing between the groups. Use different colors or patterns to improve clarity.
Ensure that units are indicated where applicable.
The titles for each subplot (A, B, C) should be more descriptive. For example, “Chemerin Levels in CSF by Oligoclonal Band Status”.
A detailed legend explaining the groups and any abbreviations used is necessary.
How was the inflammation markers assessed?
Figure 6: Correlation Analyses
Ensure that correlation coefficients are clearly indicated on the plots.
Ensure that units are indicated where applicable.
The titles for each subplot (A, B, C, D, E, F) should be more descriptive. For example, “Correlation Between Chemerin Levels in Serum and BMI”.
Were the correlations adjusted for potential confounders?
Technical Flaws in Table 2
The unit for the CSF/serum ratio should be consistently written as “× 10⁻³” instead of “x10-3”.
There are inconsistencies in the number of decimal places used. For example, “Chemerin serum [ng/mL]” is given as “297.71 + 113.26” while “Chemerin CSF [ng/mL]” is “27.80 + 8.69”. Consistency should be maintained throughout the table.
The ranges for some parameters are not clearly defined. For example, “Chemerin CSF/serum ratio x10-3” has a range of “[38.73-221.18]”, which seems unusually high and might indicate an error.
The abbreviations for low-density lipoprotein and high-density lipoprotein should be defined in the table legend or footnotes for clarity.
In the Discussion Section:
While the article compares its findings with previous studies, it does not always critically evaluate discrepancies. For example, it notes differences in chemerin levels between this study and others but does not explore potential reasons for these differences in depth.
The discussion could benefit from more mechanistic insights. For instance, the potential pathways through which chemerin affects neurological diseases are not explored in detail.
Some sentences are long and complex, making them difficult to read. For example, “The present study investigated circulating and CSF chemerin levels in paired samples from a large clinical cohort comprising patients suffering from various neurological diseases such as MS, ID, epilepsy, CVD and PC as well as from a control group of individuals without evidence of neurological diseases” could be split into two sentences for clarity.
There are missing commas in several places, such as after introductory phrases (e.g., “Interestingly, chemerin serum levels might be influenced by diet since they are reduced by oral lipid ingestion”).
The study mentions excluding BMI as a confounding variable for sex differences in chemerin levels, but it does not address other potential confounders comprehensively.
The discussion sometimes overstates the implications of the findings. For example, the statement “Our data present a solid basis for the development of standard values for chemerin quantities in CSF” is strong, given the study’s limitations.
While the article compares its findings with previous studies, it does not always critically evaluate discrepancies. For example, it notes differences in chemerin levels between this study and others but does not explore potential reasons for these differences in depth.
The discussion could benefit from more mechanistic insights. For instance, the potential pathways through which chemerin affects neurological diseases are not explored in detail.
In the conclusion:
The conclusion makes broad statements that are not fully supported by the data. For example, “Future studies should address decreased chemerin levels in MS/ID and increased levels in epilepsy” is a valid suggestion, but the article should also acknowledge the study’s limitations and the need for further validation.
While the article suggests future research directions, it could be more specific about the types of studies needed to confirm and extend the findings.
Overall, the article provides valuable data on chemerin levels in cerebrospinal fluid and their association with various neurological conditions. However, it could be improved by addressing the technical flaws, providing more detailed statistical analysis, and offering a more critical discussion of the results. Additionally, clearer presentation of data and more specific future research directions would enhance the article’s impact.
Please remove the plagiarism/similarity index of the manuscript which is above 21% and also some AI content though detected very low to make it more precise.
Comments on the Quality of English Language
Moderate editing if English is required along with technical faults elimination.
Author Response
Reviewer 2
Review Report of manuscript ID Number” Biomedicines- 3224902”
The research article entitled “Systematic quantification of chemerin in human cerebrospinal fluid” submitted for publication in “Biomedicnes” with manuscript ID Number” Biomedicines-3224902” has good healthy and comprehensive results and the manuscript is well designed and presented and contains all the necessary experiments to prove his hypothesis.
My general comments on the manuscript are incorporated as follows:
The title should be capitalized correctly: “Systematic Quantification of Chemerin in Human Cerebrospinal Fluid”
We want to thank the reviewer for this hint. We corrected the capitalisation. (p.1, ll. 2-3)
In the abstract there are several grammatical and textual mistakes such as
The unit for the CSF/serum ratio should be consistently written as “× 10⁻³” instead of “x10-3”
There are inconsistencies in the number of decimal places used such as “96.3+36.8 x10-3” should be “96.3 ± 36.8 × 10⁻³”.
The term “enzyme-linked-immunosorbent assay” should be hyphenated correctly as “enzyme-linked immunosorbent assay (ELISA)”.
The abbreviations for body mass index (BMI) and C-reactive protein (CRP) should be defined when first mentioned.
The phrase “chemerins ability” should be “chemerin’s ability”
Some sentences are long and complex, making them difficult to read such as “Dysregulation of adipokines and inflammation of adipose tissue are considered as key mechanisms of chronic inflammation in metabolic syndrome (metaflammation)” could be split into two sentences for clarity.
There are missing commas in several places, such as after introductory phrases (e.g., “Interestingly, chemerin serum levels might be influenced by diet since they are reduced by oral lipid ingestion”).
In the abstract “Data on chemerin in human cerebrospinal fluid (CSF) is sparse. Little is known about chemerins ability to cross the blood/CSF barrier under physiological and pathophysiological conditions” could be revised to “Data on chemerin in human cerebrospinal fluid (CSF) is sparse, and its ability to cross the blood/CSF barrier under various conditions is not well understood”.
Some statements in the abstract are too predetermine such as “Our data present a solid basis for the development of standard values for chemerin quantities in CSF” are too strong given the study’s limitations.
We thank the reviewer for the thoroughly revision of the abstract. We corrected the abstract as recommended. Furthermore, the abstract was edited as suggested by other reviewers (p. 1, ll. 10-24)
In the introduction:
The sentence “Obesity-related diseases are thought to be caused by dysregulated adipokine production” could be revised to “It is thought that obesity-related diseases are caused by dysregulated adipokine production”.
The sentence “In rodents, intracerebral chemerin infusion promotes food intake and expression of chemerin receptor was found in hippocampal tissue” could be split into two sentences for clarity.
There are missing commas in several places, such as after introductory phrases (e.g., “Interestingly, chemerin serum levels might be influenced by diet since they are reduced by oral lipid ingestion”).
The units“BMI ≥ 25kg/m2” should have a space between the number and the unit: “BMI ≥ 25 kg/m²”.
Some references are incomplete or incorrectly formatted. For example, should be formatted consistently with other references.
Thank you very much for these helpful comments. We corrected the addressed issues as suggested and ensured references are complete.
Results:
It is mentioned that the Mann-Whitney U-test and Kruskal-Wallis tests were used for statistical analysis. However, it does not provide sufficient detail on the assumptions of these tests.
Thank you for this comment. The samples we investigated are independent and non-parametric. Therefore, the applicable test is the Mann-Whitney U-test for 2 variables and the Kruskal-Wallis test for more than 2 variables. We added an explanation in the methods section. (p.3, ll. 135-136)
The results section also presents a lot of data in text form, which can be organized into more tables and figures to summarize key findings to improve readability.
We thank the reviewer for this comment. In the revised version of the manuscript, we ensured that all data presented in the text are displayed in tables and figures.
The study mentions excluding BMI as a confounding variable for sex differences in chemerin levels, but it does not address other potential confounders comprehensively.
Thank you for this important remark. The most important findings in correlation analyses are displayed in Figure 3. As recommended, we further investigated these results regarding potential confounders (age, BMI, CRP, gender). The results are addressed briefly in the results section and additionally provided in a supplementary table 3. (p.9, ll. 281-285)
In Table 1
The unit for BMI should be written as “kg/m²” instead of “kg/m2”.
Unfortunately, there must have occurred some error in the display of the manuscript. In our version we write kg/m². We are going to make sure that it will be written correctly in the final version.
The unit for cell count should be “cells/µL” instead of “uL”.
Thank for this hint. We corrected this error. (p.4, table 1)
The CSF/serum albumin ratio should be written as “CSF/serum albumin ratio × 10⁻³” instead of “CSF/serum Albumin ratio x10-3”.
Once again, we are sorry about the incorrect display. In our version we write 10-3 (superscripted). We are going to make sure that it will be written correctly in the final version.
There are inconsistencies in the number of decimal places used. For example, “BMI (kg/m²)” is given as “27.19 + 6.31” while “Total protein (g/L)” is “70.65 + 6.13”. Consistency should be maintained throughout the table.
We thank you for this suggestion. The number of decimal places displayed represents the number of decimal places in the original data, which are determined by the assays used to quantify the concentrations.
The ranges for some parameters are not clearly defined. For example, “Total albumin (g/L)” has a range of “[22.8-414.0]”, which seems unusually high and might indicate an error.
We thank you for your thorough revision of the manuscript. Unfortunately, there was a typo in decimal places in 1 albumin concentration. Thanks to you, we were able to correct this error and we recalculated all statistical analyses regarding albumin concentrations.
The abbreviations for low-density lipoprotein and high-density lipoprotein should be defined in the table legend or footnotes for clarity.
Parameters should be grouped logically. For example, anthropometric parameters (age, weight, height, BMI) should be grouped together, followed by serum parameters, and then CSF parameters.
Adding subgroup headings (e.g., “Anthropometric Parameters”, “Serum Parameters”, “CSF Parameters”) would improve readability.
Anthropometric Parameters should be listed first, followed by serum parameters, and then CSF parameters. This logical flow helps in understanding the data better.
The neurological subgroups should be listed at the end of the table to maintain a clear distinction between general parameters and specific subgroups.
The table legend should provide a detailed description of the parameters, including units and any abbreviations used. For example, “LDL: low-density lipoprotein; HDL: high-density lipoprotein; CRP: C-reactive protein; GOT: glutamic oxaloacetic transaminase; GPT: glutamic-pyruvic transaminase; CSF: cerebrospinal fluid; LDH: lactate dehydrogenase; Ig: immunoglobulin”.
Footnotes should be used to clarify any unusual values or ranges. For example, if the range for “Total albumin (g/L)” is correct, a footnote should explain why it is so high.
We thank the reviewer for these recommendations. We reformatted table 1 as suggested and ensured that all abbreviations are explained. (pp 4-5, ll. 160-166)
Figure 1: Neurological Diseases
The figure lacks clear labels for each subgroup. Ensure that each subgroup is clearly labeled and distinguishable. The spelling of infectious disease is incorrect.
The legend is missing or insufficient. A detailed legend explaining the subgroups and any abbreviations used is necessary.
The title should be more descriptive. For example, “Distribution of Neurological Diagnoses Among Study Participants”.
How were the subgroups determined? Were there any specific criteria?
Thank you for this hint. We elaborate the figure and the figure legend as suggested. Colours label each subgroup. To ensure clarity, we indicate labelling additionally with matching lines. The neurological diagnosis was provided by a board-certified neurologist applying standard criteria (patient history, blood analyses, CSF analyses, partially MRI). (p.6, ll. 184-186)
Figure 2: Chemerin Levels in Serum and CSF
Ensure that units are consistently used and clearly indicated on the axes.
The p-values are indicated, but the method of calculation should be mentioned in the figure legend.
The titles for each subplot (A, B, C, D) should be more descriptive. For example, “Chemerin Levels in Serum by BMI” instead of just “Serum Chemerin [ng/ml]”.
We thank the reviewer for these suggestions. We corrected the notation on the y axis in figure 2C and added a heading to figure 2. (p.6., l. 202)
Were the p-values adjusted for multiple comparisons?
We used correction for multiple comparisons whenever there was multiple comparison within one compartment (e.g. serum or CSF). Multiple comparison was defined as more than 2 subgroups. (e.g. figure 4 and 5C). We now clearly state this in the figure legends. (p.6, l. 206; p.8, l. 241; p. 8, l. 264)
Figure 3: Blood/Brain Barrier Dysfunction
The figure is not very clear in distinguishing between the groups. Use different colors or patterns to improve clarity.
Ensure that units are indicated where applicable.
The title should be more descriptive. For example, “Prevalence of Blood/Brain Barrier Dysfunction by Gender”. A detailed legend explaining the groups and any abbreviations used is necessary.
How was blood/brain barrier dysfunction assessed?
We thank the reviewer for this comment. Similar colours mark similar subgroups. We changed colours to beige for patients without a blood/brain barrier dysfunction, and rust red colour for patients with blood/brain barrier dysfunction. Units are not applicable because we display the absolute number of patients in each subgroup. We elaborated the figure legend for more clarity (p.6, ll. 210-213)
The blood brain barrier dysfunction was assessed at the Neurochemical Laboratory Giessen University Hospital, Germany by standard methods, e.g. CSF/serum albumin ratio and cell count. We added this information in the methods section (p.3, ll. 112-113)
Figure 4: Chemerin Levels by Neurological Diagnosis
Ensure that error bars are included to indicate variability.
The method of calculating p-values should be mentioned in the figure legend.
Thank you for this suggestion. To display data in Figure 4 (as well as figure 2 and figure 5) we used boxplot design. Boxplots display the median, lower and upper interquartile range. Whiskers mark minimum and maximum. We added this explanation in the methods section and in the figure legends. (p. 3, ll. 142-143; p.6, ll. 206-208; p. 8, ll. 242-243 and 265-267)
The titles for each subplot (A, B) should be more descriptive. For example, “Chemerin Levels in Serum by Neurological Diagnosis”.
Were the comparisons adjusted for potential confounders?
Thank you for these comments. The titles were elaborated as suggested. (p. 8, ll. 239-244)
To exclude potential confounders, we performed subgroup analyses with matched controls in section 3.4.
Figure 5: Chemerin Levels and CSF Inflammation Markers
The figure is not very clear in distinguishing between the groups. Use different colors or patterns to improve clarity.
Ensure that units are indicated where applicable.
The titles for each subplot (A, B, C) should be more descriptive. For example, “Chemerin Levels in CSF by Oligoclonal Band Status”.
A detailed legend explaining the groups and any abbreviations used is necessary.
How was the inflammation markers assessed?
Thank you for these suggestions. We revised the figures and used different colours to improve clarity. We ensured that all applicable units are indicated. We revised the figure legend as suggested. (p. 8, ll. 259-267)
Inflammation markers were assessed by the Neurochemical Laboratory Giessen University Hospital, Germany by standard methods (oligoclonal bands: isoelectric focusing and immunofixation; cell count: counting cells in CSF per µl; grade of blood/brain barrier: CSF/serum albumin ratio)
Figure 6: Correlation Analyses
Ensure that correlation coefficients are clearly indicated on the plots.
Ensure that units are indicated where applicable.
The titles for each subplot (A, B, C, D, E, F) should be more descriptive. For example, “Correlation Between Chemerin Levels in Serum and BMI”.
Thank you for these remarks. We adjusted the titles for each subplot as you suggested. We ensured all units are indicated.
Were the correlations adjusted for potential confounders?
Thank you for this helpful suggestion. We conducted further statistical analysis to exclude confounders in our most import findings in correlation analyses. Interestingly, gender seem to have an relevant effect on chemerin in CSF. We now mention these findings in the results section and added the results in the supplementary file. (p.9, ll. 281-285 and supplementary table 3)
Technical Flaws in Table 2
The unit for the CSF/serum ratio should be consistently written as “× 10⁻³” instead of “x10-3”.
There are inconsistencies in the number of decimal places used. For example, “Chemerin serum [ng/mL]” is given as “297.71 + 113.26” while “Chemerin CSF [ng/mL]” is “27.80 + 8.69”. Consistency should be maintained throughout the table.
The ranges for some parameters are not clearly defined. For example, “Chemerin CSF/serum ratio x10-3” has a range of “[38.73-221.18]”, which seems unusually high and might indicate an error.
The abbreviations for low-density lipoprotein and high-density lipoprotein should be defined in the table legend or footnotes for clarity.
Thank you for these remarks. Table 2 was thoroughly revised ensuring 2 decimal places each. Chemerin CSF/serum ratios are displayed correctly. Every abbreviation used in this table is defined in the table legend.
In the Discussion Section:
While the article compares its findings with previous studies, it does not always critically evaluate discrepancies. For example, it notes differences in chemerin levels between this study and others but does not explore potential reasons for these differences in depth.
The discussion could benefit from more mechanistic insights. For instance, the potential pathways through which chemerin affects neurological diseases are not explored in detail.
Some sentences are long and complex, making them difficult to read. For example, “The present study investigated circulating and CSF chemerin levels in paired samples from a large clinical cohort comprising patients suffering from various neurological diseases such as MS, ID, epilepsy, CVD and PC as well as from a control group of individuals without evidence of neurological diseases” could be split into two sentences for clarity.
There are missing commas in several places, such as after introductory phrases (e.g., “Interestingly, chemerin serum levels might be influenced by diet since they are reduced by oral lipid ingestion”).
The study mentions excluding BMI as a confounding variable for sex differences in chemerin levels, but it does not address other potential confounders comprehensively.
The discussion sometimes overstates the implications of the findings. For example, the statement “Our data present a solid basis for the development of standard values for chemerin quantities in CSF” is strong, given the study’s limitations.
While the article compares its findings with previous studies, it does not always critically evaluate discrepancies. For example, it notes differences in chemerin levels between this study and others but does not explore potential reasons for these differences in depth.
The discussion could benefit from more mechanistic insights. For instance, the potential pathways through which chemerin affects neurological diseases are not explored in detail.
In the conclusion:
The conclusion makes broad statements that are not fully supported by the data. For example, “Future studies should address decreased chemerin levels in MS/ID and increased levels in epilepsy” is a valid suggestion, but the article should also acknowledge the study’s limitations and the need for further validation.
While the article suggests future research directions, it could be more specific about the types of studies needed to confirm and extend the findings.
Overall, the article provides valuable data on chemerin levels in cerebrospinal fluid and their association with various neurological conditions. However, it could be improved by addressing the technical flaws, providing more detailed statistical analysis, and offering a more critical discussion of the results. Additionally, clearer presentation of data and more specific future research directions would enhance the article’s impact.
We thank the reviewer for this helpful advice. We revised the discussion as suggested. For example, we included mechanistical hypotheses and more concrete future directions as well as limitations of our study. e.g. (p. 12-13, ll. 381-394; p. 13, ll. 405-410, p. 13, ll. 414-420, p. 14, ll. 448-456, p. 14, ll. 471-478, p. 14, ll. 485-492)
Please remove the plagiarism/similarity index of the manuscript which is above 21% and also some AI content though detected very low to make it more precise.
We ensured that all statements are correctly cited. No AI methods were applied in the writing of this manuscript.
Reviewer 3 Report
Comments and Suggestions for Authors
Systematic quantification of chemerin in human cerebrospinal fluid by Alexandra Höpfinger et al
Dear Editor-in-Chief (Biomedicines, MDPI)
This is a very interestiing study. For the first time in literature, these authors showed a predictve value of chemerin levels as marker of disease in serum and CSF in a large and well-characterized study cohort. They provide a solid basis for standard values of chemerin in CSF in patients with and without neurological diseases.
The content of introduction is good but seems a telegram. Please, connect sentences in this part.
I would recomend its publication after replying these considerations.
Coments to the authors and editor
Overweight patients exhibited higher chemerin levels in serum and in CSF. Please, expalin the reason and signalling pathways involved in chemerin effects.
Chemerin CSF levels were higher in men. Shall you indicate if testoterone can upregulate chemerin expression in men? What factor explain the sexodimorphic expression of chemerin?
Explain signalling pathways by which chemerin promote these effects in the introduction.
Shall you justify the differential regulation (decrease or increase) on chemerin levels, depending of the neurological disease? For me it is not easy to explain why chemerin levels are upregulated by infectious disease or multiple sclerosis but reduced by epilepsy (or without effect). Is there any specific signaling pathway activated by chemerin?
Explain the molecular mechanism by which chemerin cross BBA?
Explain the translational meaning of chemerin CSF/serum ratio as diagnosis value in certain neurological diseases? What does mean low or high rates in epilepsy, inflamatory states , epilepsy., CVD. In additon, explain who obesity can affect this ratio and the contribution to the progression of neurological diseases?
Explain the conexion between chemerin levels and adipokines in neurodegenerative diseases in the introduction. Althoug the contento f introduction is clear, it seems a telegram. Please, connect parragrahs.
Since at chemerin is a chemoattractant for macrophages and dendritic cell at inflamamtion sites, explain the relevence of monocyte migration in your study as well as the posible relationship between chemerin reducton or upregulation, depènding of neurological disease in your study.
Shall you explain how chemerin donwregulation could be affected by diet in your study?
If chemerin might modulate food intake via central pathways, why chemerin is upregulated by obesity in your study? Is there comorbilites that synergically can reduce chemerin levels in your study?
Since multiple sclerosis (MS) raised chemerin levels as potential link between disease severity progression in the CNS, shall you confirm if chemerine promote monocyte recruitment in the brain in your study? Is there any published study in rodent models of MS? How contirbute obesity to enhance chemerin levels in MS patients? Is there strong systemic inflammation (cytokines) by obesity in these MS patients as compare to MS alone?
Can be consider the chemerin CSF/serum ratio as a marker of inflammation in patients with obesity? Is there connexion between MCP-1, chemerin and obesity? Can chemerin provoke apoptosis in the hypothalamus of MS patients with obesity?
How do you select their grade of blood/brain barrier dysfunction by dividing patients into subgroups (normal, slight, moderate, severe dysfunction of the blood/brain barrier)?
What do you mean ¨pseudotumor cerebri¨?
The detection range of the ELISA kit was 31.2 – 2000 pg/mL. Please, shall you indicate how standard curve were made, includin concentration of all chemerine recombinant point in the stardar curve? Indicate the sensitivy of this Duo-Set ELISA kit. How plasma samples were isolated and also indicate the compostion of used buffer for ELISA detection in the CSF
Mean values ± standard deviation (SD) were measured. I thing is more appropiate the expression of chemerin as mean values+- SEM (Estándar error media=variance/root of n, being n the size sample). I would recomed you express chemerine results as mean+- SEM. So, I would recomment you express all data as mean+-SEM.
How is posible to distinguish the contribution of meningitis or encephalitis to neurodegeneratioin and the possible relationship with chemerine levels in patients with neurological diseases? Shall you indicate the exactly cerebrovascular disease evaluated in your patients?
How do you explain this significantly higher prevalence of impaired blood/brain barrier dysfunction in men than women with lower chemerin levels?
The figures are clear but the pdf does of figure 4c is absent. Please, include this figure 4c
How you exclude the contribution of age to chemerin levels in patients with MSC and obesity? . In fact, Chemerin serum levels correlated positively with BMI (p < 0.001; rho = +0.414; n = 170, Figure 6A) and CRP levels (p < 0.001; rho = +0.570; n = 166) (Figure 6B). In addition, chemerin levels in serum and CSF showed a strong positive correlation (p = 0.008; rho = +0.201; n = 170) (Figure 6C) and the specific CSF/serum-ratio for chemerin, which positively correlated with age.
Are chemerin upregulated levels influenced by age or neurodegeneration in these patients with different neurological diseases?
The last correlation of 3.3 point does not appaer in the PDF (see figure E,F, point 3.3).
They also mention that the observed chemerin CSF/serum ratio of 96.3 x10-3 is comparable to C1q/TNF-related protein-3 (CTRP-3) CSF/serum ratio (110 x10-3 ) [27], remarkable higher compared to leptin CSF/serum ratio 3.9 x10-3 341 [28] (3.8 x10-3 [29]) or resistin CSF/serum ratio (4.4 x10-3 ) [12] but, interestingly, lower than Meteorin-Like Protein (METRNL) CSF/serum ratio (1400 x10-3 [30]).
Shall you explain how inflammation and monocyte recruitment could differentially contribute to upregulate chemerine levels and also explain these correlations with detail in the discussion.
If chemerine correlates with age, obesity in certain neurological diseases, shall you explain why chererim levels were significantly lower in MS-obesity patients when compared to those suffering MS alon or those with epilepsy or CVD? It is really difficult to assume that chemerin can increase or decrease depending of each disease. Is the age of controls different as compare their respective neurological disease you compare chemerine in your study ?
Furthermore, the CSF/serum ratio was significantly lower in patients suffering from infectious diseases. Consistent with this finding, the subset of patients with elevated cell count in CSF (≥ 5/µL) exhibited significantly lower chemerin levels and CSF chemerin levels were negatively correlated with cell count in general. Shall you explain these discrepances in terms of inflamamtion or monocyte recruitment in you study? I miss the discussion of signalling pathways associated to chemerin effects in your study. Shall you describe the signalling pathwya by which chemerin increase TNF Alpha levels in you study.
These authors suggest a role of reactive microglia in the pathogenesis of epilepsy, including phagocytosis and remodeling of the epileptic brain microenvironment [41; in fact, they suggest that chemerin contribute to the pathogenesis of epilepsy. In my opinion, these microglia overactivation also occurs in these neurological diseases and it is not restricted to epilepsy. So, discuss the micrglia overactivation in the context of neuronall loss chemokine dependent levels in all evaluted diseases. IN fact, infection also activates microglia and amplifies chemokine released in the brain (in general, including chemerin relerase).l
In the CSF, chemerin levels were significantly elevated whereas serum quantities did not change as comprare to controls. So, explain the potential value of chemerin as diagnosis in epilepsy since their levels are different in CSF and plasma.
Although the mechanisms still remain to be elucidated and the current data, I would like to highligth that neuroinflammation also occurs in case of infection diseases and CVD. Thus, microglia overactation in not only exclusive of epilpetic patients. Please, discuss some published findings in rodent models of disease with microglia and chemokine in the context of neuronal loss and compare findings with your results in different neurological diseases.
Author Response
Reviewer 3
Systematic quantification of chemerin in human cerebrospinal fluid by Alexandra Höpfinger et al
Dear Editor-in-Chief (Biomedicines, MDPI)
This is a very interestiing study. For the first time in literature, these authors showed a predictve value of chemerin levels as marker of disease in serum and CSF in a large and well-characterized study cohort. They provide a solid basis for standard values of chemerin in CSF in patients with and without neurological diseases.
The content of introduction is good but seems a telegram. Please, connect sentences in this part.
Thank you for this suggestion. We rephrased parts of the introduction section as suggested.
I would recomend its publication after replying these considerations.
Coments to the authors and editor
Overweight patients exhibited higher chemerin levels in serum and in CSF. Please, expalin the reason and signalling pathways involved in chemerin effects.
We thank you for this comment. We added a paragraph on this topic in the discussion section. (p. 13, ll. 397-402.
“Chemerin is strongly expressed in adipocytes and induced during adipocyte differentiation, reaching its apex in mature adipocytes [36]. Chemerin appears to play an au-tocrine/paracrine role in white adipose tissue [37]. It is able to affect recruitment of macrophages in white adipose tissue via inhibition of matrix metalloproteinase 3 and chemokines via NFkB signalling [37]. Endogenous chemerin can increase chemerin activity via an autocrine positive feedback.”
Chemerin CSF levels were higher in men. Shall you indicate if testoterone can upregulate chemerin expression in men? What factor explain the sexodimorphic expression of chemerin? Explain signalling pathways by which chemerin promote these effects in the introduction.
We thank the reviewer for this remark. We elaborated the discussion section to discuss this issue. (pp.12-13, ll. 387-394)
“A previous study has shown that testosterone inhibits chemerin secretion in murine adipocytes in-vitro [21]. Therefore, increased chemerin in CSF in male patients is more likely caused by surrounding factors (e.g. dysfunctional blood/brain barrier, disease-related factors) than direct regulation via androgens. Nonetheless, chemerin receptors have been found in Leydig cells of the male reproductive tract and chemerin was suggested as a regulator in male gonadal steroidogenesis [35]. Future studies should investigate effects of central chemerin on reproductive functions.”
Shall you justify the differential regulation (decrease or increase) on chemerin levels, depending of the neurological disease? For me it is not easy to explain why chemerin levels are upregulated by infectious disease or multiple sclerosis but reduced by epilepsy (or without effect). Is there any specific signaling pathway activated by chemerin?
We thank the reviewer for this excellent question. In the entire study, patients suffering from MS or infectious CNS disease show lower chemerin levels in CSF compared to patients suffering from CVD. At this point, we can only speculate the reasons of differential regulation on chemerin levels depending on the neurological disease.
In a murine stroke model, recombinant chemerin was reported to exert protective effects on neuronal and blood/brain barrier damage. Therefore, elevated chemerin levels in CSF might contribute to repair mechanisms in the brain. We elaborated the discussion section on this topic (p. 14, ll. 485-492).
In matched control subgroups, patients suffering from infectious diseases show lower chemerin CSF levels. Known chemerin receptors are G protein-coupled receptors CMKLR1 (chemokine-like receptor 1) and GPR1(G protein-coupled receptor 1). Activation of these chemerin receptors lead to intracellular signalling cascades resulting in activation or inhibition of kinases. Chemerin induces the production of proinflammatory cytokines like IL-6, IL-8, and TNF-α (DOI:10.1155/2017/5468023). An intracellular pathway involving STAT (Signal transducer and activator of transcription) 3 signalling is assumed (DOI: 10.1007/s11033-024-09359-y). We added information about signalling of chemerin in the introduction section and elaborated the discussion section regarding proposed pathways activated by chemerin. (p.2, ll. 50-53; p. 14, ll. 448-453)
“In a murine model of autoimmune encephalomyelitis, the chemerin receptor CMKLR1 (chemokine-like receptor-1) is expressed in microglial cells and CNS-infiltrating myeloid dendritic cells. In vitro, chemerin is able to trigger β-arrestin-2 association with CMKLR1 and induce cell migration of these CMKLR1+ cells. Antagonizing these chemerin effects suppressed CNS autoimmune inflammation in a murine model of autoimmune encephalomyelitis [42]. Moreover, chemerin receptor deficient mice exhibited less severe disease phenotypes. Therefore, chemerin appears to play a pivotal role in development of MS, at least in mice.”
At this point, we can only speculate about reasons of reduced chemerin levels in CSF in infectious diseases. We elaborated the discussion section on this topic. (p. 14, ll. 471-478)
“… these findings might indicate infection-driven down-regulation of either autochthonous chemerin expression specifically in the CNS or of chemerin permeability through the blood/CSF barrier. Nonetheless, chemerin expression is increased in an inflammatory environment by TNF-α [46], yet proteolytic procession of chemerin was found at site of inflammation [47] and might result in decreased chemerin levels in CSF in inflammatory diseases. Future studies should test these hypotheses and elucidate the relevance and mechanisms of reduced chemerin levels in infectious CNS diseases..”
Explain the molecular mechanism by which chemerin cross BBA?
The molecular mechanisms by which chemerin crosses the blood/brain barrier are unknown and have not been investigated so far. Future studies should focus on this subject. We added this suggestion in the discussion section. (p. 13, ll. 414-420). Chemerin could be actively transported via the blood brain barrier. For the best-characterized adipokine leptin, a saturable, specific, temperature-dependent receptor was found at the human blood-brain barrier (Golden et al. J Clin Invest. 1997 Jan 1;99(1):14-8. doi: 10.1172/JCI119125). To the best of our knowledge, a similar receptor for chemerin has not been investigated so far.
“…mechanisms of chemerin crossing the blood/brain barrier have not been investigated so far. For the best researched adipokine leptin, a saturable, specific, temperature-dependent receptor was found at the human blood/brain barrier [39]. In our study, obese patients had elevated CSF/serum ratios. This might support the hypothesis that chemerin levels in CSF depend, at least in part, on chemerin serum concentrations. Future studies should investigate molecular mechanisms of chemerin crossing the blood/brain barrier under physiological and pathophysiological conditions in vivo.”
Explain the translational meaning of chemerin CSF/serum ratio as diagnosis value in certain neurological diseases? What does mean low or high rates in epilepsy, inflamatory states , epilepsy., CVD. In additon, explain who obesity can affect this ratio and the contribution to the progression of neurological diseases?
We thank the reviewer for these suggestions. We elaborated the discussion section on this topic, for example:
“In clinical practice, CSF/serum ratios are of high relevance. Especially CSF/serum ratios of albumin and immunoglobulins are used for diagnostic purposes [28]. Increased ratios are either caused by impaired blood/brain barrier function or by an increased autochthonous production in the CNS. Reduced CSF/serum ratios can be due to increased serum levels and stable blood/brain barrier permeability, e.g. if the substance is actively transported via the blood/brain barrier independent of serum concentrations.” (p. 12, ll. 354-360)
“In our study, obese patients had elevated CSF/serum ratios. This might support the hypothesis that chemerin levels in CSF depend, at least in part, on chemerin serum concentrations.” (p. 13, ll. 416-418)
Explain the conexion between chemerin levels and adipokines in neurodegenerative diseases in the introduction. Althoug the contento f introduction is clear, it seems a telegram. Please, connect parragrahs.
We thank the reviewer for this recommendation. We connected paragraphs as suggested in the introduction section. Furthermore, we explain the link between adipokines in neurodegenerative diseases. To the best of our knowledge, chemerin levels have not been investigated in patients with neurodegenerative diseases. In our study, there are no patients with neurodegenerative diseases included.
“Via the so-called fat-brain axis, adipokines might affect brain metabolism, brain atrophy, cognitive decline and neuronal inflammation” (p.1, ll. 33-35)
Since at chemerin is a chemoattractant for macrophages and dendritic cell at inflamamtion sites, explain the relevence of monocyte migration in your study as well as the posible relationship between chemerin reducton or upregulation, depènding of neurological disease in your study.
We thank you for this important and valid point. Since chemerin acts on monocyte/microglia migration, it is reasonable to assume, that this capacity plays an important role in neurological diseases. In a murine mouse model of autoimmune encephalomyelitis for MS, activation of CMKLR1 (a chemerin receptor) by chemerin is crucial for migration of microglia and CNS-infiltrating myeloid dendritic cells and disease progression (doi:10.4049/jimmunol.0803435, doi:10.1371/journal.pone.0112925). Furthermore, in a murine ischemic stroke model, chemerin reduced microglial inflammatory response and neuronal apoptosis. We elaborated the discussion section on these topics. (p. 14, ll. 448-456, 485-492)
Shall you explain how chemerin donwregulation could be affected by diet in your study?
This is truly an important remark. We added a section “limitations” in our manuscript. (p. 15, ll. 519-523).
“A previous study has shown that chemerin serum levels are reduced after oral lipid ingestion [21]. Due to the retrospective nature of the present study, we have no information about patients eating habits prior to sampling of serum and CSF. Therefore, short-term effects of ingested lipids cannot be excluded.”
If chemerin might modulate food intake via central pathways, why chemerin is upregulated by obesity in your study? Is there comorbilites that synergically can reduce chemerin levels in your study?
It is known, that adipocytes in white adipose tissue secrete chemerin. Furthermore, endogen chemerin induces endogen chemerin secretion via an autocrine positive feedback loop. Therefore, chemerin levels in serum are associated with increased BMI. In our study, we are able to demonstrate a parallel observation in CSF. We elaborated the discussion section on this topic. (p. 13, ll. 405-410)
“Different hypotheses can be drawn from this observation: increased chemerin levels in serum could lead to proportionally increased chemerin levels in CSF. Furthermore, the blood/brain barrier could be more permeable for chemerin in obese patients. An in-creased autochthone chemerin production in the central nervous system in obese patients also cannot be excluded due to our study design. Future studies should be de-signed to test these hypotheses.”
Since multiple sclerosis (MS) raised chemerin levels as potential link between disease severity progression in the CNS, shall you confirm if chemerine promote monocyte recruitment in the brain in your study? Is there any published study in rodent models of MS? How contirbute obesity to enhance chemerin levels in MS patients? Is there strong systemic inflammation (cytokines) by obesity in these MS patients as compare to MS alone?
The potential link of chemerin and MS is intruiging. We elaborated the discussion section on this topic. (p. 14, ll. 448-456)
“In a murine model of autoimmune encephalomyelitis, the chemerin receptor CMKLR1 (chemokine-like receptor-1) is expressed in microglial cells and CNS-infiltrating myeloid dendritic cells [26]. In vitro, chemerin is able to trigger β-arrestin-2 association with CMKLR1 and induce cell migration of these CMKLR1+ cells [43]. Antagonizing these chemerin effects suppressed CNS autoimmune inflammation in a murine model of autoimmune encephalomyelitis [43]. Moreover, chemerin receptor deficient mice exhibited less severe disease phenotypes [26]. Therefore, chemerin appears to play a pivotal role in development of MS, at least in mice. Future studies should focus on the role of chemerin in patients suffering from MS. “
In this study, patients suffering from MS with a BMI less than 25 kg/m² showed lower chemerin levels in serum than patients with an BMI of 25 kg/m² or higher. Chemerin levels in CNS did not differ in patients suffering from MS regarding their BMI subgroup.
In our study, there are only CSF and serum samples available. Therefore, we cannot investigate monocyte recruitment in human brain. Nonetheless, this question should be adressed in future studies. In our study, cell count in CSF, CRP and leukocytes are avaiable as surrogate parameters for inflammation (table 1, figure 4, figure 6). In MS patients, obese patients did not show higher CRP-levels or cell count in CSF than non-obese patients. In the future, other inflammatory markers such as cytokines should be investigated.
Can be consider the chemerin CSF/serum ratio as a marker of inflammation in patients with obesity? Is there connexion between MCP-1, chemerin and obesity? Can chemerin provoke apoptosis in the hypothalamus of MS patients with obesity?
We thank you for these excellent questions. In our study, inoverweight patients, CSF/serum ratio of chemerin is not positively correlated with markers of inflammation like systemic CRP, systemic leukocyte count or cell count in CSF. Therefore, our current data do not suggest CSF/serum ratio of chemerin as a marker of inflammation in obese patients.
Both, MCP-1 and chemerin are produced by adipocytes (DOI: 10.1007/s10620-008-0585-3) and serum levels of both adipkines are increased in patients suffering von type 2 diabetes mellitus compared to patients with normal glucose regulation (doi: 10.17305/bjbms.2019.4002). Interestingly, chemerin and MCP-1 secretion are both reduced by the anti-inflammatory omega-3-fatty acid docosahexaenoic acid in human adipocytes (DOI: 10.1186/s12986-016-0064-3). Therefore, related regulation mechanisms can be assumed.
Your question about chemerin-triggered apoptosis in the hypothalamus of MS pastients in obesity is really intruiging. Unfortunately, due to the cross-sectional retrospective nature of our study, we are not able to answer this question at present, but it should be adressed in future studies.
How do you select their grade of blood/brain barrier dysfunction by dividing patients into subgroups (normal, slight, moderate, severe dysfunction of the blood/brain barrier)?
The grade of blood brain/barrier dysfunction was assessed at the Neurochemical Laboratory Giessen University Hospital, Germany by standard methods, e.g. CSF/serum albumin ratio and cell count in CSF. We added this information in the methods section. (p.3, ll. 112-113)
What do you mean ¨pseudotumor cerebri¨?
Pseudotumor cerebri is a condition also called idiopathic intracranial hypertension. We added this in the methods section (p.3, l. 118-119). Diagnostic criteria are an elevated CSF pressure more than 25 cm H2O, exclusion of pathological CSF cell count, exclusion of structural or vascular lesion in MRI.
The detection range of the ELISA kit was 31.2 – 2000 pg/mL. Please, shall you indicate how standard curve were made, includin concentration of all chemerine recombinant point in the stardar curve? Indicate the sensitivy of this Duo-Set ELISA kit. How plasma samples were isolated and also indicate the compostion of used buffer for ELISA detection in the CSF
Standard curve was made according to the instructions suggested by the selling company: a series of dilutions with factor 2. (e.i. 2000, 1000, 500, etc) was conducted. The dilution buffer 1% BSA in PBS was used for serum and CSF samples as well as chemerin standards. We added this information in the methods section. (p.3, ll. 127-131)
Mean values ± standard deviation (SD) were measured. I thing is more appropiate the expression of chemerin as mean values+- SEM (Estándar error media=variance/root of n, being n the size sample). I would recomed you express chemerine results as mean+- SEM. So, I would recomment you express all data as mean+-SEM.
Thank you very much for this suggestion. In literature about chemerin, other authors used mean values ± standard deviation. Especially in studies, where chemerin in CSF was investigated, other authors displayed mean values ± standard deviation. To improve comparabilty between our study and other studies in this field of research, we would prefer mean values ± standard deviation. We hope, that you agree with us on this point. If you do not agree, please let us know. (e.g.: Zhao et al, J Biol Chem. 2011 Nov 11; 286(45): 39520–39527. doi: 10.1074/jbc.M111.258954; Tomalka-Kochanowska et al Neuro Endocrinol Lett. 2014;35(3):218-23.)
How is posible to distinguish the contribution of meningitis or encephalitis to neurodegeneratioin and the possible relationship with chemerine levels in patients with neurological diseases? Shall you indicate the exactly cerebrovascular disease evaluated in your patients?
This study is a cross-sectional retrospective study. Patients were diagnosed by a board-certified neurologist by patients’ history, presentations of symptoms, blood work, CSF results and in some cases MRI. Due to the number of patient samples, patients with infectious diseases (e.g. meningitis, encephalitis) were summarized in 1 group. Patients suffering from stroke in differing cerebral areas and transient cerebral ischaemic attacks were summarized in 1 group referred to as cerebrovascular diseases. We added this information in the methods section. (p.3, ll. 117-118)
How do you explain this significantly higher prevalence of impaired blood/brain barrier dysfunction in men than women with lower chemerin levels?
We thank the reviewer for this important question. This was a retrospective study and there was no selection of patients prior to statistical analyses. Therefore, gender is not equally distributed in subgroups of neurological diseases. Impaired blood/brain barrier is found in patients suffering from infectious diseases. In this subgroup, there were 43.4 % male and 56.7 % female patients. Among patients without neurological diseases, there were only 34.1 % male and 65.9 % female patients. Patients suffering from pseudotumor cerebri are mostly in female. In our study, there were 80.8 % female and only 19.2 % male patients. In general, these patients do not show an impaired blood/brain barrier. Therefore, a significant gender bias is caused by the unequally gender distribution in neurological diseases. We added this in the discussion section.
“In our retrospective study as well as in real life, gender is not equally distributed among neurological diseases. For example, infectious diseases with impaired blood/brain barrier occur in similar proportion in male and female patients. Patients suffering from pseudotumor cerebri, who rarely show impaired blood/brain barrier function, are more likely to be female. These gender related differences regarding neurological pathologies may contribute to the significant difference of chemerin levels in CSF by gender.” (p.12, ll. 381-387)
The figures are clear but the pdf does of figure 4c is absent. Please, include this figure 4c
Figure 4 consists of panels 4A and 4B: Chemerin levels in serum by neurological diagnosis (A) and chemerin levels in CSF by neurological diagnosis (B).
How you exclude the contribution of age to chemerin levels in patients with MSC and obesity? . In fact, Chemerin serum levels correlated positively with BMI (p < 0.001; rho = +0.414; n = 170, Figure 6A) and CRP levels (p < 0.001; rho = +0.570; n = 166) (Figure 6B). In addition, chemerin levels in serum and CSF showed a strong positive correlation (p = 0.008; rho = +0.201; n = 170) (Figure 6C) and the specific CSF/serum-ratio for chemerin, which positively correlated with age.
Thank you for this remark. To exclude age as a confounding variable in our correlation analyses, we added partial correlation analyses in the revised version of the manuscript. We were able to exclude age a confounding variabel for the correlation of chemerin levels and BMI. In the revised version of the manuscript, we mention this finding now in the results section (p. 9, ll. 281-285) and added a supplemetary table 3 at the end of the revised manuscript.
The last correlation of 3.3 point does not appaer in the PDF (see figure E,F, point 3.3).
We are sorry for this error in your PDF version. We will ensure that all figures will appear in the revised version.
They also mention that the observed chemerin CSF/serum ratio of 96.3 x10-3 is comparable to C1q/TNF-related protein-3 (CTRP-3) CSF/serum ratio (110 x10-3 ) [27], remarkable higher compared to leptin CSF/serum ratio 3.9 x10-3 341 [28] (3.8 x10-3 [29]) or resistin CSF/serum ratio (4.4 x10-3 ) [12] but, interestingly, lower than Meteorin-Like Protein (METRNL) CSF/serum ratio (1400 x10-3 [30]). Shall you explain how inflammation and monocyte recruitment could differentially contribute to upregulate chemerine levels and also explain these correlations with detail in the discussion.
We thank the reviewer for this comment. We elaborated the discussion section on the topic.
“These adipokines were recently described in CSF, suggesting them as putative media-tors of the fat-brain axis. However, for most adipokines, mechanisms by which they cross the blood/barrier are unknown so far.” (p.12, ll. 364-367)
“…. In vitro, chemerin is able to trigger β-arrestin-2 association with CMKLR1 and induce cell migration of these CMKLR1+ cells” (p. 14, ll. 450-451)
“Nonetheless, chemerin expression is increased in an inflammatory environment by TNF-α [46], yet proteolytic procession of chemerin was found at site of inflammation [47] and might result in decreased chemerin levels in CSF in inflammatory diseases.” (p. 14, ll. 473-476)
If chemerine correlates with age, obesity in certain neurological diseases, shall you explain why chererim levels were significantly lower in MS-obesity patients when compared to those suffering MS alon or those with epilepsy or CVD? It is really difficult to assume that chemerin can increase or decrease depending of each disease. Is the age of controls different as compare their respective neurological disease you compare chemerine in your study ?
Thank you for this remark. In patients suffering from MS with a BMI lower than 25 kg/m² mean value of chemerin in serum was 267.37 ± 99.13 ng/mL and in CSF 27.99 ± 8.22 ng/ml. In MS patients with a BMI of 25 kg/m² or higher, chemerin levels in serum was 321.66 ± 119.13 ng/mL, in CSF 28.44 ±9.10 ng/mL
In section 3.4 and associated table 2, we conducted subgroup analyses for each disease group. We matched a control for each patient regarding sex, age and BMI.
Furthermore, the CSF/serum ratio was significantly lower in patients suffering from infectious diseases. Consistent with this finding, the subset of patients with elevated cell count in CSF (≥ 5/µL) exhibited significantly lower chemerin levels and CSF chemerin levels were negatively correlated with cell count in general. Shall you explain these discrepances in terms of inflamamtion or monocyte recruitment in you study? I miss the discussion of signalling pathways associated to chemerin effects in your study. Shall you describe the signalling pathwya by which chemerin increase TNF Alpha levels in you study.
We thank the reviewer for this comment. In the revised version of the manuscript, we elaborated the discussion section on this topic thanks to your input. For example:
“Chemerin is strongly expressed in adipocytes and induced during adipocyte differentiation, reaching its apex in mature adipocytes [36]. Chemerin appears to play an autocrine/paracrine role in white adipose tissue [37]. It is able to affect recruitment of macrophages in white adipose tissue via inhibition of matrix metalloproteinase 3 and chemokines via NFkB signalling [37]. Endogenous chemerin can increase chemerin activity via an autocrine positive feedback.” (p. 13, ll. 397-402)
“In a murine model of autoimmune encephalomyelitis, the chemerin receptor CMKLR1 (chemokine-like receptor-1) is expressed in microglial cells and CNS-infiltrating mye-loid dendritic cells [26]. In vitro, chemerin is able to trigger β-arrestin-2 association with CMKLR1 and induce cell migration of these CMKLR1+ cells [43]. Antagonizing these chemerin effects suppressed CNS autoimmune inflammation in a murine model of autoimmune encephalomyelitis.” (p.14, ll. 448-453)
“Chemerin is a well-known modulator in the innate immune system and increases the release of proinflammatory cytokines like tumor necrosis factor (TNF)-α or inter-leukin (IL)-6 [44] via G protein-coupled receptors. “ (p. 14, ll. 457-459)
“Nonetheless, chemerin expression is increased in an inflammatory environment by TNF-α [46], yet proteolytic procession of chemerin was found at site of inflammation [47] and might result in decreased chemerin levels in CSF in inflammatory diseases.” (p. 14, ll. 473-478)
These authors suggest a role of reactive microglia in the pathogenesis of epilepsy, including phagocytosis and remodeling of the epileptic brain microenvironment [41; in fact, they suggest that chemerin contribute to the pathogenesis of epilepsy. In my opinion, these microglia overactivation also occurs in these neurological diseases and it is not restricted to epilepsy. So, discuss the micrglia overactivation in the context of neuronall loss chemokine dependent levels in all evaluted diseases. IN fact, infection also activates microglia and amplifies chemokine released in the brain (in general, including chemerin relerase). In the CSF, chemerin levels were significantly elevated whereas serum quantities did not change as comprare to controls. So, explain the potential value of chemerin as diagnosis in epilepsy since their levels are different in CSF and plasma.
We thank the reviewer for this important comment. In fact, microglia activation and migration play an important role in many CNS diseases, such as MS, infectious diseases, CVD and epilepsy. As mentioned above, chemerin acts directly on microglia via CMKLR1 and leads to microglia migration.
We elaborated the discussion section on this topic for example:
“In a murine stroke model, recombinant chemerin was reported to exert protective effects on neuronal and blood/brain barrier damage [50]. Furthermore, chemerin reduces microglial inflammatory response and neuronal apoptosis [51]. Therefore, an upregulation of chemerin in CVD patients can be speculated to reduce neuronal and blood/brain barrier damage. Future studies should investigate chemerin’s impact on neuronal damage in CVD patients.” (p. 14, ll. 486-492)
To the best of our knowledge, chemerin so far has been investigated in only 3 different studies in the context of epilepsy. Chyrea et al. investigated the effect of ketogenic diet on chemerin serum levels in children with epilepsy (PMID: 35490359). Elhady et al. observed a positive correlation between chemerin serum levels and seizure severity (DOI: 10.1007/s10072-018-3448-5). But both studies were lacking CSF samples for analyses of chemerin CSF levels or mechanistical experiments. The third study investigated expression chemerin receptor in hippocampal tissue, which was increased after status epilepticus (DOI: 10.1093/brain/awy247). We elaborated the discussion section on this topic.
“Therefore, chemerin levels in CSF might represent a new diagnostic parameter in patients suffering from epilepsy. Future studies should investigate chemerin CSF levels in larger cohorts of epileptic patients, with further focus on disease severity. Interestingly, increased serum chemerin levels were also observed in idiopathic epilepsy and chemerin concentrations were associated with seizure severity as was reported by a previous study including 50 children [53]. However, unfortunately, this study was lacking CSF samples [53]. Expression of chemerin receptor was found in hippocampal tissue after status epilepticus [23], but corresponding function of chemerin receptor in hippocampal tissue has not been investigated so far. Underlying mechanisms of chemerin in epilepsy remain to be elucidated and the current data provide a solid basis for future mechanistical studies focusing on this intriguing issue.” (p. 15, ll. 498-508)
Although the mechanisms still remain to be elucidated and the current data, I would like to highligth that neuroinflammation also occurs in case of infection diseases and CVD. Thus, microglia overactation in not only exclusive of epilpetic patients. Please, discuss some published findings in rodent models of disease with microglia and chemokine in the context of neuronal loss and compare findings with your results in different neurological diseases.
We thank the reviewer for this important remark. As mentioned above, we elaborated the discussion section on this important topic. (p. 14, ll. 448-456; p. 14, ll. 485-492)
Round 2
Reviewer 1 Report
Comments and Suggestions for Authors.